# Dynamic Fair Division with Partial Information

**Gerdus Benadè**
Boston University
benade@bu.edu

**Daniel Halpern**
Harvard University
dhalpern@g.harvard.edu

**Alexandros Psomas**
Purdue University
apsomas@cs.perdue.edu

## Abstract

We consider the fundamental problem of fairly and efficiently allocating $T$ indivisible items among $n$ agents with additive preferences. The items become available over a sequence of rounds, and every item must be allocated immediately and irrevocably before the next one arrives. Previous work shows that when the agents' valuations for the items are drawn from known distributions, it is possible (under mild technical assumptions) to find allocations that are envy-free with high probability and Pareto efficient ex-post.

We study a *partial-information* setting, where it is possible to elicit ordinal but not cardinal information. When a new item arrives, the algorithm can query each agent for the relative rank of this item with respect to a subset of the past items. When values are drawn from i.i.d. distributions, we give an algorithm that is envy-free and $(1-\epsilon)$-welfare-maximizing with high probability. We provide similar guarantees (envy-freeness and a constant approximation to welfare with high probability) even with minimally expressive queries that ask for a comparison to a single previous item. For independent but non-identical agents, we obtain envy-freeness and a constant approximation to Pareto efficiency with high probability. We prove that all our results are asymptotically tight.

## 1 Introduction

We consider the following fundamental problem in fair division. A set of $T$ indivisible items, arriving one at a time, must be allocated among a set of $n$ agents with additive preferences. The value $v_{i,t}$ that agent $i$ has for the item in round $t$ is realized once the item arrives. Each item must be allocated immediately and irrevocably upon arrival, and we ask that the overall allocation is *fair* and *efficient*.

Previous work on this problem shows that, despite the uncertainty about future valuations, one can achieve simultaneous fairness and efficiency when agents' values are stochastic. Specifically, when each $v_{i,t}$ is drawn i.i.d. from a distribution $D$, the simple algorithm that maximizes welfare — each item is allocated to the agent with the highest value — is envy-free with high probability and (obviously) ex-post Pareto efficient [DGK$^+$14, KPW16]. The same guarantee holds for independent and non-identical agents ($v_{i,t}$ is drawn from an agent-specific distribution $D_i$) for the algorithm that maximizes weighted welfare [BG22]. Even when agents' valuations for an item are correlated (but items are independent), Pareto efficiency ex-post is compatible with strong fairness guarantees ("envy-freeness with high probability or envy-freeness up-to-one item ex-post") [ZP20].

Despite the computational simplicity of (most of) the aforementioned algorithms, an unappealing aspect, especially from a practical perspective, is the requirement that agents report an exact numerical value for each item. Eliciting expressive additive valuations can be impractical, e.g., due to agents' cognitive limitations. Motivated by such considerations, a growing body of work in AI studies what can be achieved by algorithms that only elicit *ordinal* information. This idea originates from [PR06], who defined the notion of distortion to measure the worst-case deterioration of an aggregate cardinal objective (e.g., utilitarian social welfare) due to only having access to preferences of limited expressiveness, particularly ordinal rankings. Recent works

36th Conference on Neural Information Processing Systems (NeurIPS 2022).

prove bounds on the distortion in the context of many problems in social choice, e.g., voting [CNPS17, GKM17, MSW20, MW19, Kem20a, Kem20b, GHS20], matching [FRFZ14], and participatory budgeting [BNPS21]; see [AFRSV21] for a recent survey.

In this paper, we study the power and limits of eliciting ordinal information in dynamic fair division. The value $v_{i,t}$ of agent $i$ for item $t$ is drawn from an *unknown* distribution upon arrival, and the algorithm is provided, from each agent, partial ordinal information about this item, e.g., its rank relative to the past items allocated to this agent, or even just a single past item allocated to this agent. Under what distributional assumptions and elicitation constraints, can we simultaneously achieve qualitative fairness and efficiency? We answer these questions.

## 1.1 Our Contribution

We start by establishing a separation between the cardinal setting and our ordinal one. Pareto efficiency alone is trivial (allocate all goods to the same agent) and, in the cardinal setting, it is known that Pareto efficiency ex-post is compatible with envy-freeness with high probability (as long as agents are independent). We prove (Theorem 4) that in our setting, even for the case of two i.i.d. agents and any *known* distribution, envy-freeness with high probability is incompatible with even a very mild notion of (exact) Pareto efficiency, one-swap-Pareto efficiency, which requires that there is no beneficial one-to-one trade of items between agents (but allows for improvements via many-to-many trades of items).

We proceed to give an essentially matching positive result. For any number of i.i.d. agents and an unknown distribution $D$, there exists an algorithm (Algorithm 1) that is envy-free with high probability and guarantees a $(1-\varepsilon)$-approximation to the optimal utilitarian social welfare (the sum of utilities), for all $\varepsilon > 0$, with high probability (Theorem 5). When an item arrives, the algorithm learns for each agent $i$ its relative rank compared to a subset of prior items allocated to agent $i$ (but otherwise knows nothing about the underlying numerical valuation). Our algorithm works in epochs. Each epoch has an exploration/sampling phase, where each agent gets a pre-determined number of items, followed by an exploitation/ranking phase, where each fresh item is given to an agent whose empirical quantile is largest. The goal is to make a sublinear number of errors compared to the "ideal" algorithm that allocates each item to the agent with the highest true quantile. The algorithm has to balance the need for sampling, which leads to more accurate empirical quantiles, against the number of inefficient allocations made while sampling. A significant technical barrier is that we cannot fix a target accuracy because the underlying distribution is unknown. That is, for every fixed accuracy for the empirical quantiles, there exists a distribution for which this accuracy is not good enough for even a constant approximation to the optimal welfare. Instead, we need to make our epochs progressively longer, thereby guaranteeing progressively better bounds.

Given this strong positive result, we explore the limits of what we can achieve when further restricting the amount of information available. What if each agent can remember only a *single* item previously allocated to them, and the fresh item is compared to just this one item?[1] Surprisingly, the aforementioned positive result can almost be recovered even in this very restrictive setting. We prove that there exists an algorithm (Algorithm 2) that is envy-free with high probability and guarantees a $(1 - 1/e)^2 - \varepsilon$ approximation to the optimal welfare with high probability, for all $\varepsilon > 0$ (Theorem 9). Our algorithm again proceeds in epochs with gradually increasing exploration and exploitation phases; this time the goal is a sublinear number of differences compared to allocating each item to a uniformly random agent with quantile at least $1 - 1/n$, which we prove is envy-free and approximately efficient (Lemmas 2 and 1). When exploring, the algorithm puts a new item in memory, estimates its quantile, and rejects it if not sufficiently close to $1 - 1/n$. We need to sample enough to ensure high confidence in the estimated quantile, but also account for the additional sampling since an item's quantile value might be far from $1-1/n$ to begin with; several technical details need to be accounted for. We give a near-matching lower bound: no algorithm with a memory of one item can achieve a $0.999-$approximation to the social welfare with high probability; therefore a constant approximation (which Algorithm 2 provides) is all we can hope for.

Finally, we relax the i.i.d. condition and study agents that are independent but not identical; each agent $i$'s values are drawn from an unknown distribution $D_i$. Even with unbounded memory, we

---

[1]So, the algorithm only learns if the new item is better or worse than the item in memory and may, at that time, choose to replace the item in memory.

Table 1: Main Results

| Setting | Possibility | Impossibility |
|---|---|---|
| Unbounded Memory, i.i.d. | EF + $(1-\varepsilon)$-welfare (Theorem 5) | EF + PO (Theorem 4) |
| Limited Memory, i.i.d. | EF + .399-welfare (Theorem 9) | .999-welfare (Theorem 8) |
| Unbounded Memory, non-i.i.d. | EF + .367-PO (Theorem 13) | EF + .809-PO (Theorem 12) |

show that it is impossible to get a $\frac{1+\sqrt{5}}{4} \approx .809$ approximation to Pareto efficiency with probability more than $2/3$, even for two agents (Theorem 12). At the same time, we prove that Algorithms 1 and 2 are envy-free and $1/e$ approximately Pareto efficient with high probability! Note that, though both algorithms give the same formal guarantees and Algorithm 2 elicits strictly less information, one might still prefer to use Algorithm 1 since it has significantly shorter exploration phases. A summary of our results can be found in Table 1

We leave the study of correlated agents as an interesting open problem. Finally, we note that beyond stochastic valuations, [BKPP18] show that it is possible to achieve sublinear envy by randomly allocating every item when agents' valuations are adversarially generated (and this is optimal); however, sublinear envy is incompatible with non-trivial efficiency even in the cardinal setting [ZP20].

## 1.2 Related Work

A number of works study fair division under ordinal preferences, e.g., [AGMW15, BEL10, BBL+17, NNR17], but often these models do not assume an underlying cardinal model and work directly on the ordinal preferences. [ABM16] assume underlying cardinal information and, among other results, bound the approximation ratio of truthful mechanisms that elicit rankings. Closer to our work, [HS21] study rules that have access to the ranking of the top-$k$ items of each agent and bound the ratio of the social welfare of the allocation returned by a rule in the worst case. They also characterize the value of $k$ needed to achieve prominent notions of fairness, namely envy-freeness up to one item (EF1) and approximate maximin share guarantee (MMS), as well as bound the loss in efficiency incurred due to fairness constraints in this setting.

Our work contributes to the growing literature in dynamic fair division [KPS14, AAGW15, FPV15, FPV17, BKPP18, LLL18, HPPZ19, ZP20, GPT21, BKM22, GBI21, VPF21, BGGJ22, BMS22, KS22] (and we note that the welfare-maximizing algorithms of [DGK+14, KPW16, BG22] work in the dynamic setting, even though the their settings are not explicitly dynamic). To the best of our knowledge, we are the first to study imperfect expressivity in a dynamic setting in fair division.

## 2 Preliminaries

A set of $T$ indivisible items/goods, labeled by $\mathcal{G} = \{1, 2, \cdots, T\}$, needs to be allocated to a set of $n$ agents, labeled by $\mathcal{N} = \{1, \ldots, n\}$. Agent $i \in \mathcal{N}$ assigns a value $v_{i,t} \in [0, 1]$ to item $t \in \mathcal{G}$. We assume agents have *additive* valuation functions, so $v_i(S) = \sum_{t \in S} v_{i,t}$ for $S \subseteq \mathcal{G}$. An allocation $A$ is a partition of the items into bundles $A_1, \ldots, A_n$, where $A_i$ is the set of items assigned to agent $i \in \mathcal{N}$. Each allocation has an associated utility profile $v(A) = (v_1(A_1), \ldots, v_n(A_n))$.

Items arrive online, one per round. The agents' valuations for the item in round $t$ (the $t$-th item) are realized when the item arrives. Every item is allocated immediately and irrevocably before moving on to the next round. We write $\mathcal{G}^t = \{1, 2, \cdots, t\}$ for the set of items that arrived in the first $t$ rounds, and $A_i^t$ for the allocation of agent $i$ after the $t$-th item was allocated. We consider two different models which specify how values are generated. In the **i.i.d. model**, the value of agent $i$ for item $t$ is independently drawn from an *unknown* distribution $D$ with CDF $F$, i.e., $v_{i,t} \sim D$ for all $i \in \mathcal{N}$ and $t \in \mathcal{G}$. In the **non-i.i.d. model**, the value of item $t$ for agent $i$ is independently drawn from an *unknown*, agent-dependent distribution $D_i$ with CDF $F_i$, i.e., $v_{i,t} \sim D_i$ for all $i \in \mathcal{N}$ and $t \in \mathcal{G}$. We write $X_i$ for the random variable for $i$'s valuation, and $X_{i,t}$ for the random variable for $i$'s valuation for item $t$. It is often convenient to work directly with the quantile of an agent's value rather than the value itself; let $Q_i = F_i(X_i)$ and $Q_{i,t} = F_i(X_{i,t})$ respectively be the random variable denoting the quantile of agent $i$ the associated item. Note that all $Q_i$ and $Q_{i,t}$ are i.i.d. and

follow a Unif$[0, 1]$ distribution. Unless explicitly stated otherwise, we assume all distributions are continuous (i.e., do not have point masses).

**Ordinal information.** We assume the realizations $v_{i,t}$ are not available. Instead, our algorithms have access to *ordinal* information. Specifically, given current item $t$, the algorithm can access each agent's *ranking* for $S = \{t\} \cup M$, $M \subseteq \mathcal{G}^{t-1}$. The size of $M$, which we will informally refer to as the *memory size*, determines the complexity of eliciting information from each agent. In one extreme, agent $i$ compares a new item $t$ to a single item they had previously received, i.e., $M \subseteq A_i^{t-1}, |M| \leq 1$. In the other extreme, $t$ is compared to all previous items she received, so $M = A_i^{t-1}$. We write $\sigma_i(S)$ for the ranking of agent $i$ for a subset $S$ of the items, and $\sigma_i^{-1}(S, t)$ for the position of item $t \in S$ with respect to a subset $S$ according to agent $i$. The highest value item is in position 1 and the lowest in position $|S|$. For example, if $S = \{1, 4\}$, $v_{i,1} = 1$ and $v_{i,4} = 0.1$, $\sigma_i(S) = (1 \succ 4)$, $\sigma_i^{-1}(S, 1) = 1$ and $\sigma_i^{-1}(S, 4) = 2$.

**Algorithms.** An algorithm $\mathcal{A}$, in each step $t$, queries each agent for ordinal information with respect to some subset $M$ and then makes a (possibly randomized) allocation decision based on this ordinal information and the history so far. An instance of our problem is parameterized by the number of agents $n$ and the (unknown) value distributions $D_1, \ldots, D_n$. Let $\mathcal{E}_P(t)$ be the event that some algorithm satisfies property P (e.g., envy-freeness or PO or $\varepsilon$-welfare) at time $t$. We are interested in the probability that an algorithm satisfies certain properties in the limit, as the number of rounds tends to infinity, where the randomness is over the random choices of the algorithm as well as the randomness in the valuations.

**Definition 1.** An algorithm satisfies $P$ with high probability if $\lim_{t \to \infty} \Pr[\mathcal{E}_P(t)] = 1$.

Note that this definition of high probability allows for dependency on $n$ and the underlying distributions (i.e., they are treated as constants).

**Efficiency notions.** An allocation $A$ Pareto dominates an allocation $A'$, denoted $A \succ A'$, when $v_i(A_i) \geq v_i(A_i')$ for all $i \in \mathcal{N}$ and there exists $j \in \mathcal{N}$ with $v_j(A_j) > v_j(A_j')$. An allocation $A$ is *Pareto efficient* or *Pareto optimal* (PO) if there is no feasible (integral) allocation that Pareto dominates it. An allocation $A'$ is in the (one) swap-neighborhood of $A$ when it can be created from $A$ with a single exchange of items between one pair of agents. Formally, there exist $i, j \in \mathcal{N}$ and items $z_j \in A_j$ and $z_i \in A_i$ so that $A_i' = (A_i \setminus \{z_i\}) \cup \{z_j\}$, $A_j' = (A_j \setminus \{z_j\}) \cup \{z_i\}$, and $A_k' = A_k$ for all other agents $k \neq i, j$. An allocation $A$ is *one-swap Pareto optimal* (SPO) if it is undominated by any allocation in its swap-neighborhood. We use a notion of approximate efficiency defined by [RF90] according to which an allocation $A$ is *$\alpha$-Pareto efficient* when $v(A)/\alpha$ is undominated.

The social welfare of an allocation $A$ is $\mathsf{sw}(A) = \sum_{i \in \mathcal{N}} v_i(A_i)$. Let allocation $A^*$ denote a (social) welfare optimal allocation for which $\mathsf{sw}(A^*) \geq \mathsf{sw}(A)$ for all feasible allocations $A$. An allocation provides an $\alpha$-approximation to welfare if $\mathsf{sw}(A) \geq \alpha \cdot \mathsf{sw}(A^*)$.

**Fairness notions.** We focus on a prominent notion of fairness called *envy-freeness*. An allocation $A^T = (A_1^T, \ldots, A_n^T)$ of $T$ items is *envy-free* (EF) when $v_i(A_i^T) \geq v_i(A_j^T)$ for all $i, j \in \mathcal{N}$, and $c$-strongly-envy-free ($c$-strong-EF) when $v_i(A_i) \geq v_i(A_j) + cT$.

## 3 Ideal Quantile-based Algorithms.

For our analysis, it will be useful to compare our algorithms with ideal algorithms that know *exact* quantile values for every item (and, in fact, several of our lower bounds apply to these stronger algorithms too). Two ideal algorithms of interest are (1) quantile maximization, which allocates each item to the agent with the highest quantile value for it, and (2) "$q$−threshold," which allocates each item uniformly at random among agents whose quantile is at least $q$ (and uniformly at random over all agents, if all quantile values are less than $q$).

In the i.i.d. model, quantile maximization is the same as value maximization, and thus envy-free with high probability and ex-post welfare optimal. The property we will use is $c$-strong envy-freeness, for some distribution-dependent constant $c$, which we state as Lemma 1. This was essentially proved by [DGK$^+$14]; we provide an alternate proof that also works, essentially unchanged, for the $\frac{n-1}{n}$-threshold algorithm; it can be found in Appendix A.1. The main idea is that an agent's value for

an item conditioned on them receiving it is strictly larger than their value for an item received by another agent. Combined with the fact that the probability of receiving an item is equal for all agents, standard concentration results ensure each agent strictly prefers their own bundle with high probability.

**Lemma 1.** *[Essentially [DGK+14].] In the i.i.d. and non-i.i.d. models, the quantile maximization algorithm and the $\frac{n-1}{n}$-threshold algorithm are $c$-strongly-envy-free, with high probability, where the constant $c = \min_{i \in \mathcal{N}}(\mathbb{E}[X_i \mid Q_i \geq 1/2] - \mathbb{E}[X_i])/(4n)$.*

Note that $c$ is strictly positive since our distributions are continuous. In the i.i.d. model, we show that the $\frac{n-1}{n}$-threshold algorithm gives a $\left(1 - \frac{1}{e}\right)^2 - \varepsilon$ approximation to welfare (Lemma 2) with high probability.

Next, we prove that the $\frac{n-1}{n}$-threshold algorithm guarnatees a constant approximation to welfare. The main idea for this proof is that, with constant probability, *some* agent has a quantile of at least $\frac{n-1}{n}$ for each item. Additionally, giving an item to an agent with quantile at least $\frac{n-1}{n}$ generates within a constant factor amount of welfare as giving it to the agent with the highest welfare.

**Lemma 2.** *In the i.i.d. model, the $\frac{n-1}{n}$-threshold algorithm guarantees a $\left(\left(1 - \frac{1}{e}\right)^2 - \varepsilon\right)$-approximation to welfare, with high probability, for all $\varepsilon > 0$.*

*Proof.* Let $F$ be the CDF of an arbitrary continuous distribution. The expected contribution of an item to the welfare of the threshold algorithm is at least

$$\mathbb{E}_{Q \sim \text{Unif}[0,1]}\left[F^{-1}(Q) \mid Q \geq \frac{n-1}{n}\right] \cdot \Pr_{\vec{Q} \sim \text{Unif}[0,1]^n}\left[\max_{i \in \mathcal{N}} Q_i \geq \frac{n-1}{n}\right].$$

For the first term we have

$$\mathbb{E}_{Q \sim \text{Unif}[0,1]}\left[F^{-1}(Q) \mid Q \geq \frac{n-1}{n}\right] = \mathbb{E}_{Q \sim \text{Unif}[\frac{n-1}{n},1]}\left[F^{-1}(Q)\right]$$

$$= \left(\int_{\frac{n-1}{n}}^1 F^{-1}(q) \cdot f_{\text{Unif}[\frac{n-1}{n},1]}(q) \, dq\right)$$

$$= \left(\int_{\frac{n-1}{n}}^1 F^{-1}(q) \cdot n \, dq\right)$$

$$\geq^{(f_{\text{Beta}[n,1]}(x)=nx^{n-1})} \left(\int_{\frac{n-1}{n}}^1 F^{-1}(q) \cdot f_{\text{Beta}[n,1]}(q) \, dq\right)$$

$$= \mathbb{E}_{Q \sim \text{Beta}[n,1]}\left[F^{-1}(Q) \mid Q \geq \frac{n-1}{n}\right] \cdot \Pr_{Q \sim \text{Beta}[n,1]}\left[Q \geq \frac{n-1}{n}\right]$$

$$\geq \mathbb{E}_{Q \sim \text{Beta}[n,1]}\left[F^{-1}(Q)\right] \cdot \Pr_{Q \sim \text{Beta}[n,1]}\left[Q \geq \frac{n-1}{n}\right]$$

$$= \mathbb{E}_{\vec{Q} \sim \text{Unif}[0,1]^n}\left[F^{-1}(\max_{i \in \mathcal{N}} Q_i)\right] \cdot \Pr_{\vec{Q} \sim \text{Unif}[0,1]^n}\left[\max_{i \in \mathcal{N}} Q_i \geq \frac{n-1}{n}\right],$$

where we used the fact that the maximum of $n$ draws from $U[0,1]$ follows a $\text{Beta}(n,1)$. The expected contribution to the welfare is thus at least

$$\mathbb{E}_{\vec{Q} \sim \text{Unif}[0,1]^n}\left[F^{-1}(\max_{i \in \mathcal{N}} Q_i)\right]\left(1 - \left(1 - \frac{1}{n}\right)^n\right)^2 \geq \left(1 - \frac{1}{e}\right)^2 \mathbb{E}_{\vec{Q} \sim \text{Unif}[0,1]^n}\left[F^{-1}(\max_{i \in \mathcal{N}} Q_i)\right]$$

Finally, for any fixed $\varepsilon > 0$, standard Chernoff bounds tell us that with high probability, the optimal welfare of $T$ items is at most $T \cdot (1 + \varepsilon/2)\mathbb{E}_{\vec{Q} \sim \text{Unif}[0,1]^n}\left[F^{-1}(\max_{i \in \mathcal{N}} Q_i)\right]$ while the welfare of the threshold algorithm is at least $T \cdot (1 - \varepsilon/2)\left(1 - \frac{1}{e}\right)^2 [\mathbb{E}_{\vec{Q} \sim \text{Unif}[0,1]^n}\left[F^{-1}(\max_{i \in \mathcal{N}} Q_i)\right]$. Indeed, the expected optimal welfare is equal to $T \cdot \mathbb{E}_{\vec{Q} \sim \text{Unif}[0,1]^n}\left[F^{-1}(\max_{i \in \mathcal{N}} Q_i)\right]$, the sum of $T$ i.i.d. random variables with expectation $\mathbb{E}_{\vec{Q} \sim \text{Unif}[0,1]^n}\left[F^{-1}(\max_{i \in \mathcal{N}} Q_i)\right]$. The standard multiplicative

Chernoff bound says that the sum of i.i.d. variables exceeds $(1 + \epsilon/2)$ times its expectation $\mu$ is at most $\exp\left(-\mu\epsilon^2/12\right)$. Plugging in $\mu = T \cdot \mathbb{E}_{\vec{Q} \sim \text{Unif}[0,1]^n}\left[F^{-1}(\max_{i \in \mathcal{N}} Q_i)\right]$, we get the desired statement. The statement about the welfare of the threshold algorithm follows similarly. Thus, the algorithm is a

$$\left(1 - \frac{1}{e}\right)^2 \cdot (1 - \varepsilon/2)/(1 + \varepsilon/2) \geq \left(1 - \frac{1}{e}\right)^2 (1 - \varepsilon) \geq \left(1 - \frac{1}{e}\right)^2 - \varepsilon$$

approximation to welfare, with high probability. □

Finally, we prove that both ideal algorithms are approximately efficient. Let $\mathcal{P}^*$ be the following property of an allocation: all items such that exactly one agent has quantile values at least $1 - 1/n$ are in the bundle of this agent. Both ideal algorithms (quantile maximization and $1 - 1/n$-threshold) satisfy $\mathcal{P}^*$. We prove that, in the non-i.i.d. model, $\mathcal{P}^*$ implies an almost $1/e$ approximation to efficiency. Our proof uses the fact that there is a (roughly) $1/e$ probability that exactly one agent has the high quantile, so the value of an agent's bundle in an algorithm that satisfies $\mathcal{P}^*$ is, with high probability, a $1/e$ approximation to their value for their $T/n$ most valuable items. Therefore, when considering an alternate allocation $A'$, the agent in $A'$ that gets at most $T/n$ items cannot be improved upon by more than a $1/e$ factor. The proof can be found in Appendix A.2.

**Lemma 3.** *In the non-i.i.d. model, every algorithm whose allocations satisfy $\mathcal{P}^*$ is $(1/e - \varepsilon)$-Pareto optimal, with high probability, for all $\varepsilon > 0$.*

## 4 Unbounded Memory in the I.I.D. Model

We explore some fundamental limits of our setting. Efficiency by itself is easy: allocate all items to the same agent. However, in contrast to the cardinal setting, we find one-swap Pareto efficiency is incompatible with envy-freeness with high probability, even for two i.i.d. agents, and even when the underlying distribution is *known*.

**Theorem 4.** *In the i.i.d. model, even for $n = 2$ agents, there does not exist an algorithm $\mathcal{A}$ which is one-swap Pareto efficient and envy-free with high probability, even when values are sampled according to $D$, for any continuous, bounded and known value distribution $D$.*

The proof can be found in Appendix B.1. The main idea is the following. As the agents are a priori identical, we can assume without loss of generality that $\mathcal{A}$ gives the first item to agent 1. The proof shows that, with positive probability, this decision becomes an irrevocable "mistake," in the sense that agent 2 really liked the item and agent 1 did not. This mistake will make envy-freeness and one-swap-Pareto efficiency incompatible.

Theorem 4 implies that when we have access to only ordinal information, we need to settle for *some* approximation to envy-freeness and efficiency. Our main positive result for this section is an algorithm that essentially matches the aforementioned lower bound.

**Theorem 5.** *In the i.i.d. model, Algorithm 1 achieves envy-freeness and a $(1 - \varepsilon)$ approximation to welfare, with high probability, for all $\varepsilon > 0$.*

Algorithm 1 works in epochs: each epoch $k$ has an exploration/sampling phase, where each agent $i$ receives a pre-determined set of items, denoted $G_i^k$, irrespective of their valuation. This is followed by an exploitation/ranking phase, where each item is given to the agent with the highest empirical quantile (with respect to items received in the preceding exploration phase, i.e. $G_i^k$). The key idea for the proof is that the exploration phases will ensure nearly all of the items will go to the agent that truly has the highest quantile. Using this, the good properties of the ideal quantile maximization algorithm will carry over to Algorithm 1.

We start with a technical lemma, which gives us a bound on the length of the exploration period we need in each epoch. The following definition will be useful.

**Definition 2.** A sample of $n \cdot m$ items (where each agent is allocated exactly $m$ items) is $\varepsilon$-accurate if, with probability at least $1 - \varepsilon$, the relative rank of a fresh item (with respect to the sample) is highest for the agent with highest quantile value.

**Lemma 6.** *If $\varepsilon, \delta \in (0, 1)$, and $m \in \mathbb{Z}^+$ are such that $\varepsilon > 2n\sqrt{\frac{\ln(2n/\delta)}{2m}}$, then giving $m$ samples to each agent is $\varepsilon$-accurate with probability at least $1 - \delta$.*

**Algorithm 1:** EF + $(1 - \varepsilon)$-Welfare

---

**for** *epoch* $k = 1 \ldots$ **do**
    **Sampling Phase:** $(n \cdot k^4$ items$)$
    Give the $j$-th item in this phase to agent $j (\bmod n)$.
    **Ranking Phase:** $(k^8$ items$)$
    **for** *each item $g$ in this phase* **do**
        Elicit $\sigma_i^{-1}(G_i^k \cup \{g\}, g)$ for all $i \in \mathcal{N}$.
        Allocate $g$ to an agent $j \in \arg\min_{i \in \mathcal{N}} \sigma_i^{-1}(G_i^k \cup \{g\}, g)$.

---

*Proof.* We will use the Dvoretzky–Kiefer–Wolfowitz (DKW) inequality [DKW56, Mas90] to show the empirical CDF of sampled quantiles is reasonably close to a uniform distribution with probability $1 - \delta$. We then show this is sufficient to guarantee $\varepsilon$-accuracy for the chosen $\varepsilon$. Let $\hat{F}_i$ be the empirical CDF of the sampled *quantiles* for agent $i$, i.e., $\hat{F}_i(q)$ for $q \in [0, 1]$ is a random variable that describes the proportion of sampled items with quantile at most $q$. Note that $\hat{F}_i$ exactly captures agent $i$'s ranking for a new item: if a fresh item has quantile $q_i$ for agent $i$ and $q_j$ for agent $j$, then $i$ ranks it higher than $j$ exactly when $\hat{F}_i(q_i) > \hat{F}_j(q_j)$.

Noting that the CDF for the actual quantile distribution (i.e., the uniform distribution) is the identity on $[0, 1]$, the DKW inequality states that for all $\gamma > 0$, $\Pr\left[\sup_{q \in [0,1]} |\hat{F}_i(q) - q| > \gamma\right] \leq 2e^{-2m\gamma^2}$. We want this condition to hold for all $n$ agents, simultaneously, with probability at least $1 - \delta$, so we pick $\gamma$ such that $2e^{-2m\gamma^2} \leq \delta/n$ and apply a union bound; it suffices to choose $\gamma = \sqrt{\frac{\ln(2n/\delta)}{2m}}$.

We now show that the DKW condition $(\sup_{q \in [0,1]} |\hat{F}_i(q) - q| \leq \gamma)$ being satisfied for all agents $i$ is sufficient to guarantee $\varepsilon$-accuracy. Consider sampling quantiles $Q_1, \ldots, Q_n$ for a fresh item. Let $i^{\max} \in \arg\max_{i \in \mathcal{N}} Q_i$ be a quantile-maximizing agent (technically a random variable). Our goal is to show that with probability at least $1 - \varepsilon$ (with respect to the samples of $Q_1, \ldots, Q_n$) $\hat{F}_{i^{\max}}(Q_{i^{max}}) > \hat{F}_j(Q_j)$ for all $j \neq i^{\max}$. This ensures that $i^{\max}$ has the highest empirical rank, and hence receives the item. Let $Q_{(1)}, \ldots, Q_{(n)}$ be the respective order statistics. A key observation is that $Q_{(n)} - Q_{(n-1)} \sim \mathrm{Beta}[1, n]$ [Gen19]. The PDF of a $\mathrm{Beta}[1, n]$ distribution is $f(x) = nx^{n-1}$ for $x \in [0, 1]$. Since $f(x) \leq n$, $\Pr\left[Q_{(n)} - Q_{(n-1)} < \rho\right] < n\rho$ for all $\rho > 0$. Plugging in $\rho = 2\gamma$, we have $\Pr\left[Q_{(n)} - Q_{(n-1)} \leq 2\gamma\right] < 2n\gamma$. We will show that as long as $\varepsilon > 2n\gamma$, $\varepsilon$-accuracy holds. First, we have $\Pr\left[Q_{(n)} - Q_{(n-1)} > 2\gamma\right] > 1 - \varepsilon$. Conditioned on $Q_{(n)} - Q_{(n-1)} > 2\gamma$, the item is given to $i^{\max}$. To see why, observe $Q_{i^{\max}} = Q_{(n)}$ and $Q_j \leq Q_{(n-1)}$ for all $j \neq i^{\max}$, by definition. Using the DKW inequality condition, it follows that $\hat{F}_{i^{\max}}(Q_{i^{\max}}) \geq Q_{i^{\max}} - \gamma > Q_j + \gamma \geq \hat{F}_j(Q_j)$. We conclude that for $\varepsilon > 2n\sqrt{\frac{\ln(2n/\delta)}{2m}}$, $\varepsilon$-accuracy is satisfied with probability at least $1 - \delta$. $\qquad\square$

Using Lemma 6, we can get, for each epoch, a bound on the number of decisions where Algorithm 1 differs from the quantile maximization algorithm.

**Lemma 7.** *The allocation of Algorithm 1 differs from that of the quantile maximization algorithm after $T$ steps by at most $f(T)$ items with high probability, where $f(T) \in O(poly(n) \cdot T^{15/16})$.*

*Proof.* We start by bounding the accuracy of Algorithm 1 in each epoch $k$. In epoch $k$, each agent receives $k^4$ items during the sampling phase. We claim that the sample in epoch $k$ for $k \geq 3n$ is $\varepsilon_k$-accurate for $\varepsilon_k := 3n/k^{3/2}$ with probability at least $1 - \delta_k$, for $\delta_k := 2n/e^{2k}$. Indeed, first note that by the choice of $k$, we have that $\varepsilon_k, \delta_k \in (0, 1)$. Hence, we just need to show that these values satisfy the inequality of Lemma 6. We have that

$$\varepsilon_k = \frac{3n}{k^{3/2}} > \frac{2n}{k^{3/2}} = 2n\sqrt{\frac{1}{k^3}} = 2n\sqrt{\frac{\ln(e^{2k})}{2k^4}} = 2n\sqrt{\frac{\ln(2n/\delta_k)}{2k^4}}.$$

Next, fix a time $T$. Slightly abusing notation, let $k(t) = \min\{K \in \mathbb{N} | \sum_{k=1}^{K} nk^4 + k^8 \geq t\}$ be the function that given an item $t$ returns the epoch item $t$ is in. Notice that $T \geq \sum_{k=1}^{k(T)-1} nk^4 + k^8 \geq$

$(k(T) - 1)^8$, and therefore $k(T) \leq 2T^{1/8}$. In any run of the algorithm, we can classify every item $t \leq T$ into one of four categories.

1. Item $t$ was allocated in one of the first $3n - 1$ epochs, that is, $k(t) < 3n$.

2. Item $t$ was allocated in the sampling phase of epoch $k(t) \geq 3n$.

3. Item $t$ was allocated in the ranking phase of epoch $k(t) \geq 3n$; the epoch was $\varepsilon_{k(t)}$-accurate.

4. Item $t$ was allocated in the ranking phase of epoch $k(t) \geq 3n$; the epoch was not $\varepsilon_{k(t)}$-accurate.

We say an item $t$ was a mistake if it was given to an agent with a non-maximum quantile for it. We show that the number of mistakes in each category are bounded by $3^{10}n^9, 2nT^{5/8}, 9nT^{15/16}$, and $158n\ln(T)$ respectively, with high probability. This implies that the total number of mistakes is at most the sum of these quantities, which is $O(\text{poly}(n) \cdot T^{15/16})$, with high probability, via a union bound.

The number of items in the first category is at most

$$\sum_{k=1}^{3n-1} k^4 n + k^8 \leq \sum_{k=1}^{3n} (3n)^4 n + (3n)^8 \leq (3n)^5 n + (3n)^9 \leq 3^{10} n^9.$$

Hence, the number of mistakes in the first category is also at most $3^{10}n^9$.

For the second category, since $k(T) \leq 2T^{1/8}$, we have that the total number of items in the sampling phase is (with probability 1) upper bounded by

$$\sum_{k=1}^{k(T)} nk^4 \leq nk(T)^5 \leq 2nT^{5/8}.$$

Each item $t$ in the third category has probability $\varepsilon_{k(t)}$ of being a mistake. The expected number of mistakes is therefore at most $\sum_{k=3n}^{k(T)} \varepsilon_{k(t)} k^8 = \sum_{k=3n}^{k(T)} 3nk^{13/2} \leq 3nk(T)^{15/2} \leq 8nT^{15/16}$. Using Hoeffding's inequality we get that with high probability the number of mistakes is at most $9nT^{15/16}$, since a deviation of $nT^{15/16}$ occurs with probability at most $\exp(-2n^2T^{15/8}/T) = \exp(-2n^2T^{7/8})$.

For the fourth category, the expected number of items is at most $\sum_{k=3n}^{k(T)} \delta_k k^8 = 2n \sum_{k=3n}^{k(T)} \frac{k^8}{e^{2k}} \leq 2n \sum_{k=1}^{\infty} \frac{k^8}{e^{2k}} \leq 158n$. Using Markov's inequality we have that the number of mistakes is at most $158n\ln(T)$ with probability at least $1 - \ln(T)$, i.e., with high probability. $\qquad\square$

Finally, we can prove Theorem 5 as a relatively straightforward consequence of Lemma 7, since the ideal quantile maximization algorithm satisfies nice properties (e.g., Lemma 1). The full proof can be found in Appendix B.2.

## 5 Bounded Memory in the I.I.D. Model

In this section, we are interested in the more ambitious problem of designing dynamic algorithms with even more limited partial information: each agent is allowed to "remember" only a single item. We first show that, in this case, we need to settle for constant approximations of welfare. The main idea of the proof is a difficult value distribution. Agents have essentially one of three values, low, medium, high. The key idea is that no matter what quantile thresholds an algorithm has, it will either be the case that with reasonable probability, the highest value agent has medium value but is indistinguishable from all low value agents, or the highest value agent has high value, but is indistinguishable from all medium or low value agents. In either case, the algorithm will make a mistake with constant probability, and hence cannot guarantee close to optimal welfare. The full proof can be found in Appendix C.1.

**Theorem 8.** *In the i.i.d. model, given a memory of one item per agent, there is no algorithm $\mathcal{A}$ that is .999-welfare maximizing with high probability for all continuous and bounded value distributions.*

---

**Algorithm 2:** Bounded Memory

---

**for** *Epoch* $k = 1 \ldots$ **do**
    **Sampling Phase:** ($k^9$ items)
    NOTWITHINERROR $\leftarrow \mathcal{N}$
    **for** *trial* $= 1, \ldots, k^3$ **do**
        **for** $i \in$ NOTWITHINERROR **do**
            | Allocate the next item to agent $i$, and update her memory
        Test $k^6 - |$NOTWITHINERROR$|$ number of items (for each agent)
        **for** $i \in$ NOTWITHINERROR **do**
            **if** *Proportion of test items for agent $i$ is within $\pm 1/k^2$ of $(n-1)/n$* **then**
                | NOTWITHINERROR $\leftarrow$ NOTWITHINERROR $\setminus \{\, i \,\}$
    **Ranking Phase:** ($k^{18}$ items)
    **for** *each item $g$ in this phase* **do**
        **if** *Some agent $i$ has high signal* **then**
            | Give $g$ to a (uniformly) random such agent
        **else**
            | Give $g$ to an agent uniformly at random

---

Our positive result matches this lower bound up to a constant.

**Theorem 9.** *In the i.i.d. model, given a memory of one item per agent, Algorithm 2 achieves envy-freeness and a $(1 - 1/e)^2 - \varepsilon$ approximation to welfare, with high probability, for all $\varepsilon > 0$.*

Algorithm 2 works in epochs, similar to Algorithm 1. In each epoch's exploration/sampling phase, it tries to find an item whose quantile is close to the $\frac{n-1}{n}$-threshold algorithm. Epoch $k$ makes $k^3$ such attempts, and each candidate item is tested against $k^6$ fresh items to get an estimated quantile. If everything is within the error we can tolerate, the algorithm remembers this item for this epoch; otherwise, the agent has an arbitrary item in memory during this epoch. During the exploitation/ranking phase, Algorithm 2 tries to mimic the $\frac{n-1}{n}$-threshold algorithm (instead of the quantile maximization algorithm as Algorithm 1 did), and, in fact, inherits its approximation factor (Lemma 2) exactly.

Our first technical lemma, Lemma 10, gives necessary bounds on the various variables of Algorithm 2 for a sample to be $\varepsilon$-accurate with respect to the ideal threshold algorithm; see Definition 3. Its proof can be found in Appendix C.2.

**Definition 3.** A set of $n$ items in memory, one for each agent, is $\varepsilon$-accurate with respect to $q^*$ if with probability at least $1 - \varepsilon$, when a fresh item is sampled, the agents with true quantile above $q^*$ are exactly those that value the fresh item more than their item in memory.

**Lemma 10.** *For all $\varepsilon, \delta \in (0, 1)$, if (1) at least $\tau$ trials are done with $\tau \geq \frac{\ln(2n/\delta)}{\varepsilon/(3n)}$, and (2) at least $\ell$ test items are used per trial for $\ell \geq \frac{18n^2}{\varepsilon^2} \ln\left(\frac{4\tau n}{\delta}\right)$, and (3) the tolerance for accepting an item is $\varepsilon/(3n)$, then the items in memory are $\varepsilon$-accurate (for all agents, simultaneously) with respect to $q^* = \frac{n-1}{n}$, with probability at least $1 - \delta$.*

Though Lemmas 6 and 10 resemble each other (and are used in analogous ways), the proofs require different techniques, as the sampling processes are very different. Next, we prove an analogue to Lemma 7: the number of disagreements between Algorithm 2 and the ideal threshold algorithm is sublinear.[2] The proofs of Lemmas 7 and 11 are similar, precisely because Lemma 6 matches Lemma 10. Theorem 9 follows from Lemma 11 as in the i.i.d. case. The proofs of Lemma 11 and Theorem 9 can be found in Appendices C.3 and C.4 respectively.

**Lemma 11.** *The allocation of Algorithm 2 differs from that of the $\frac{n-1}{n}$-threshold algorithm after $T$ steps by at most $f(T)$ items with high probability, where $f(T) \in O(poly(n) \cdot T^{17/18})$.*

---

[2] Note these are randomized algorithms, so by "differ on a item" here we mean that the distributions over agents receiving the item differ.

# 6 The Non-I.I.D. Model

In this section, we study the non-i.i.d. model. We first establish a strong lower bound for the non-i.i.d. model. The following negative result holds even for algorithms that know the associated quantile for every fresh item.

**Theorem 12.** *Even for $2$ non-identical agents, there is no algorithm that is EF and $c$-PO with probability $p$, for $c > \frac{1+\sqrt{5}}{4} \approx .809$ and $p > 2/3$, for all continuous and bounded value distributions.*

Here we provide a proof sketch of Theorem 12. The full proof can be found in Appendix D.1.

*Proof Sketch.* The main idea is that an algorithm, even if it can see the exact quantiles of all items, cannot distinguish between agents having potentially different kinds of distributions. At one extreme end, agents may have nearly identical values for all items. On the other extreme, agents may only care (i.e., have high value) for, say, their top third of items.

Consider the case of two agents. Notice that if both agents have nearly identical values for all items, to be EF, the bundle sizes of both agents must be approximately the same. We can then show that, no matter how an algorithm does this, it must be the case that one agent, say agent 1, receives a significant number of items with quantile outside of their top third does not receive a significant number of items with quantile in their top third.

However, suppose instead the value distributions were such that agent 2 still has nearly identical values for all items, but agent 1 only has high value for their top-third items. In the setting described above, agent 1 would be willing to trade many of their bottom two-third quantile items for the items agent 2 received in their top third. By setting up this trade correctly, both agents total utilities go up by a significant amount. The existence of such a trade violates constant approximations to PO. □

Algorithms 1 and 2 are envy-free with high probability, even in the non-i.i.d. model, since envy-freeness is not an "inter-agent" property. Our last result shows that they also give a constant approximation to Pareto efficiency, by combining Lemma 3 with Lemmas 7 and 11. Its proof can be found in Appendix D.2.

**Theorem 13.** *In the non-i.i.d. model, both Algorithm 1 (unbounded memory) and Algorithm 2 (one-item memory) are EF and $(1/e - \varepsilon)$-PO, with high probability, for all $\varepsilon > 0$.*

Interestingly, the guarantees for Algorithm 2 in the non-i.i.d. model are only marginally worse compared to the i.i.d. model; the approximation ratio decreases from $(1 - 1/e)^2 \approx 0.4$ to $1/e \approx 0.37$. Finally, we note that, even though the formal guarantees in Theorem 13 are the same for the two algorithms, and even though Algorithm 2 uses memory size of one, Algorithm 1 has the benefit of much shorter epoch lengths (in addition to better guarantees under the i.i.d. model).

# 7 Conclusion

To conclude, we have analyzed the online fair division problem when agents only reveal partial information. In multiple settings, we have proved tight bounds on what properties are attainable. As for future work, one direction would be to move past the assumption that agent values are drawn independently, as we may not expect this to hold in practice. Additionally, there are many other forms of partial information that could potentially achieve different properties. For example, if agents can compare not just single items but small subsets of items, then it may be possible to achieve stronger results such as arbitrarily good approximations to PO even in the non-i.i.d. setting.

## Acknowledgements

Alexandros Psomas is supported in part by an NSF CAREER award CCF-2144208, a Google Research Scholar Award, and by the Algorand Centres of Excellence program managed by Algorand Foundation. Any opinions, findings, and conclusions or recommendations expressed in this material are those of the author(s) and do not necessarily reflect the views of Algorand Foundation. This material is based upon work supported by the National Science Foundation Graduate Research Fellowship Program under Grant No. DGE1745303.

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
