# Appendix

## A Proofs Missing from Section 3

### A.1 Proof of Lemma 1

We focus on quantile maximization. The same proof goes through essentially unchanged for the threshold algorithm; we explain the differences whenever it's appropriate.

Fix distributions $D_1, \ldots, D_n$ with CDFs $F_1, \ldots, F_n$. Fix two agents $i$ and $j$. We will show with high probability, $i$ does not envy $j$ (in a strong sense). Union bounding over all $\binom{n}{2}$ pairs yields the lemma statement.

As in [DGK$^+$14], we compare the expected contribution of an item $t$ to $i$'s bundle and its expected contribution to $j$'s bundle. Let $A$ be the random variable denoting the agent that received the item. We want to consider the difference $\mathbb{E}[v_{i,t} \cdot \mathbb{I}[A = i]] - \mathbb{E}[v_{i,t} \cdot \mathbb{I}[A = j]]$. Let $H_i$ be the event that $F_i(v_{i,t}) \geq \frac{n-1}{n}$, and $L_i$ be the compliment. We split each of the two terms into conditional expectations depending on the signal, beginning with the first one:

$$\mathbb{E}[v_{i,t} \cdot \mathbb{I}[A = i]] = \mathbb{E}\left[v_{i,t} \cdot \mathbb{I}[A = i] \mid H_i\right] \cdot \Pr[H_i] + \mathbb{E}\left[v_{i,t} \cdot \mathbb{I}[A = i] \mid L_i\right] \cdot \Pr[L_i].$$

Note that, under quantile maximization, $v_{i,t}$ is positively correlated with $\mathbb{I}[A = i]$: for any fixed value $v_{i,t}$, $A = i$ with probability $F(v_{i,t})^{n-1}$, which is increasing in $v_{i,t}$. Therefore, the expectation of the product is greater than or equal to the product of the expectations: $\mathbb{E}\left[v_{i,t} \cdot \mathbb{I}[A = i] \mid H_i\right] \geq \mathbb{E}\left[v_{i,t} \mid H_i\right] \cdot \Pr[A = i \mid H_i]$ and $\mathbb{E}\left[v_{i,t} \cdot \mathbb{I}[A = i] \mid L_i\right] \geq \mathbb{E}\left[v_{i,t} \mid L_i\right] \cdot \Pr[A = i \mid L_i]$. Therefore

$$\mathbb{E}[v_{i,t} \cdot \mathbb{I}[A = i]] \geq \mathbb{E}\left[v_{i,t} \mid H_i\right] \cdot \Pr[A = i \mid H_i] \cdot \Pr[H_i] + \mathbb{E}\left[v_{i,t} \mid L_i\right] \cdot \Pr[A = i \mid L_i] \cdot \Pr[L_i]$$
$$= \mathbb{E}\left[v_{i,t} \mid H_i\right] \cdot \Pr[A = i \text{ and } H_i] + \mathbb{E}\left[v_{i,t} \mid L_i\right] \cdot \Pr[A = i \text{ and } L_i].$$

For the threshold algorithm, we have equality above, since conditioned on either $H_i$ or $L_i$, $v_{i,t}$ is independent of $\mathbb{I}[A = i]$, as the allocation depends only on the high vs low signal.

On the other hand, $v_{i,t}$ is negatively correlated with $\mathbb{I}[A = j]$. Therefore

$$\mathbb{E}[v_{i,t} \cdot \mathbb{I}[A = j]] \leq \mathbb{E}\left[v_{i,t} \mid H_i\right] \cdot \Pr[A = j \text{ and } H_i] + \mathbb{E}\left[v_{i,t} \mid L_i\right] \cdot \Pr[A = j \text{ and } L_i].$$

Again, for the threshold algorithm, we have equality.

Combined, we have

$$\mathbb{E}[v_{i,t} \cdot \mathbb{I}[A = i]] - \mathbb{E}[v_{i,t} \cdot \mathbb{I}[A = j]] \geq \mathbb{E}\left[v_{i,t} \mid H_i\right] \cdot (\Pr[A = i \text{ and } H_i] - \Pr[A = j \text{ and } H_i]) \tag{1}$$

$$- \mathbb{E}\left[v_{i,t} \mid L_i\right] \cdot (\Pr[A = j \text{ and } L_i] - \Pr[A = i \text{ and } L_i]). \tag{2}$$

We analyze (1), $\Pr[A = i \text{ and } H_i] - \Pr[A = j \text{ and } H_i]$. Let $H_j$ be the event that $F_j(v_{j,t}) \geq \frac{n-1}{n}$ Let $L_j$ be its complement. We have:

$$(\Pr[A = i \text{ and } H_i \text{ and } L_j] + \Pr[A = i \text{ and } H_i \text{ and } H_j])$$
$$- (\Pr[A = j \text{ and } H_i \text{ and } L_j] + \Pr[A = j \text{ and } H_i \text{ and } H_j]).$$

Notice that $\Pr[A = j \text{ and } H_i \text{ and } L_j] = 0$ because if agent $i$ has a high quantile and $j$ has a low quantile, $j$ cannot receive the item (in either algorithm). Additionally, by symmetry, $\Pr[A = i \text{ and } H_i \text{ and } H_j] = \Pr[A = j \text{ and } H_i \text{ and } H_j]$. Therefore, (1) simplifies to $\Pr[A = i \text{ and } H_i \text{ and } L_j]$. Finally, we note that $\Pr[A = i \text{ and } H_i \text{ and } L_j] \geq \frac{1}{n-1}$, again, for both algorithms.

We analyze (2), $\Pr[A = j \text{ and } L_i] - \Pr[A = i \text{ and } L_i]$. Let $\mathcal{E}^{\text{low}}$ be the event that all agents other than $i$ have quantile lower then $\frac{n-1}{n}$. Let $\overline{\mathcal{E}^{\text{low}}}$ be its complement, the probability that at least one agent other than $i$ has a high quantile. We have:

$$\left(\Pr\left[A = j \text{ and } L_i \text{ and } \mathcal{E}^{\text{low}}\right] + \Pr\left[A = j \text{ and } L_i \text{ and } \overline{\mathcal{E}^{\text{low}}}\right]\right)$$
$$- \left(\Pr\left[A = i \text{ and } L_i \text{ and } \mathcal{E}^{\text{low}}\right] + \Pr\left[A = i \text{ and } L_i \text{ and } \overline{\mathcal{E}^{\text{low}}}\right]\right).$$

Notice that $\Pr\left[A = i \text{ and } L_i \text{ and } \overline{\mathcal{E}^{\text{low}}}\right] = 0$ because if agent $i$ has a low quantile and at least one other agent has a high quantile, $i$ cannot receive the item. Additionally, by symmetry,

$$\Pr\left[A = j \text{ and } L_i \text{ and } \mathcal{E}^{\text{low}}\right] = \Pr\left[A = i \text{ and } L_i \text{ and } \mathcal{E}^{\text{low}}\right],$$

since when all agents have low quantiles, $i$ and $j$ are equally likely to receive the item. Hence, the probability in (2) simplifies to $\Pr\left[A = j \text{ and } L_i \text{ and } \overline{\mathcal{E}^{\text{low}}}\right] = \Pr\left[A = j | L_i \text{ and } \overline{\mathcal{E}^{\text{low}}}\right] \cdot \Pr\left[L_i \text{ and } \overline{\mathcal{E}^{\text{low}}}\right]$. The first term is equal to $1/(n-1)$, since $j$ is equally likely to receive the item compared to any agent. The second term is $\frac{n-1}{n} \cdot \left(1 - \left(\frac{n-1}{n}\right)^{n-1}\right)$. Observing that $\left(\frac{n-1}{n}\right)^{n-1} = \left(1 - \frac{1}{n}\right)^{n-1} \geq \frac{1}{e}$, we have that the probability in (2) is at most $\frac{1}{n}\left(1 - \frac{1}{e}\right)$. Overall, we have shown that

$$
\begin{aligned}
\mathbb{E}[v_{i,t} \cdot \mathbb{I}[A = i]] - \mathbb{E}[v_{i,t} \cdot \mathbb{I}[A = j]] &\geq \mathbb{E}\left[v_{i,t} \mid H_i\right] \frac{1}{n-1} - \mathbb{E}\left[v_{i,t} \mid L_i\right] \frac{e-1}{en} \\
&\geq \frac{e-1}{en}\left(\mathbb{E}\left[v_{i,t} \mid H_i\right] - \mathbb{E}\left[v_{i,t} \mid L_i\right]\right) \\
&\geq \frac{1}{2n}\left(\mathbb{E}\left[v_{i,t} \mid H_i\right] - \mathbb{E}\left[v_{i,t} \mid L_i\right]\right) \\
&= \frac{1}{2n}\left(\mathbb{E}\left[X_i \mid H_i\right] - \mathbb{E}\left[X_i \mid L_i\right]\right)
\end{aligned}
$$

It remains to show that the value of $i$ for $A_i^T$ is at least her value for $A_j^T$ plus $\frac{1}{4n}\left(\mathbb{E}\left[X_i \mid H_i\right] - \mathbb{E}\left[X_i \mid L_i\right]\right)$ with high probability. Towards this, notice that the value of $i$ for $A_i^T$ minus her value for $A_j^T$ is the sum of $T$ i.i.d. random variables, supported in $[-1, 1]$, whose expectation is at least $\frac{1}{2n}\left(\mathbb{E}\left[X_i \mid H_i\right] - \mathbb{E}\left[X_i \mid L_i\right]\right)$, as we've established so far. Hoeffding's inequality then implies that the probability that this difference is less than $b = \frac{1}{4n}\left(\mathbb{E}\left[X_i \mid H_i\right] - \mathbb{E}\left[X_i \mid L_i\right]\right)$ is at most $2\exp\left(-\frac{b^2 T}{2}\right)$, i.e., exponentially small, since $b$ is a constant. Observing that $\mathbb{E}\left[X_i \mid H_i\right] \geq \mathbb{E}\left[X_i \mid Q_i \geq 1/2\right]$ and $\mathbb{E}\left[X_i \mid L_i\right] \leq \mathbb{E}[X_i]$ concludes the proof. $\qquad\square$

## A.2 Proof of Lemma 3

Fix an $\varepsilon \in (0, 1)$, and choose $\varepsilon'$ such that $\frac{1-\varepsilon'}{(1+\varepsilon')^2} \cdot \frac{1}{e} > \frac{1}{e} - \varepsilon$ (using $\varepsilon' = \varepsilon/3$ will do). Fix distributions with CDFs $F_1, \ldots, F_n$ for each agent $i \in \mathcal{N}$, and a time $T$. Suppressing the superscript, for ease of notation, let $A_i = A_i^T$ be the bundle allocated at time $T$ to each agent $i$ by an algorithm that satisfies $\mathcal{P}^*$. Let $A_i^{\text{top}}$ be the set of the $T/n$ most valuable items for each agent $i$. Let $A_i^{\text{high}} = \left\{ t \in \mathcal{G}^T \mid F_i(v_{i,t}) \geq 1 - \frac{1+\varepsilon'}{n} \right\}$ be the set of items that agent $i$ has "high" value for, in the sense that they come from the top $\frac{1+\varepsilon'}{n}$ portion of their distribution. We show the following $3n$ events, $\mathcal{E}_{ij}$ for $i \in \mathcal{N}$ and $j \in \{1, 2, 3\}$, occur simultaneously with high probability (in $T$).

1. $\mathcal{E}_{i1}$: $v_i(A_i^{\text{top}}) \leq v_i(A_i^{\text{high}})$.

2. $\mathcal{E}_{i2}$: $v_i(A_i^{\text{high}}) \leq T \cdot \frac{(1+\varepsilon')^2}{n} \mathbb{E}_{Q \sim \text{Unif}[1-1/n, 1]}[F^{-1}(Q)]$.

3. $\mathcal{E}_{i3}$: $v_i(A_i) \geq T \cdot \frac{1-\varepsilon'}{en} \mathbb{E}_{Q \sim \text{Unif}[1-1/n, 1]}[F^{-1}(Q)]$.

Each of these individually will follow from a straightforward application of Hoeffding's inequality or Chernoff bounds, showing they each individually occur with probability exponentially close to 1 in $T$. This implies that they all occur simultaneously with high probability. Finally, we will show that conditioned on all $3n$ occurring, the allocation is $(1/e - \varepsilon)$-PO.

Let us begin with $\mathcal{E}_{i1}$ for each agent $i$. The event occurs when there are at least $T/n$ items $t \in \mathcal{G}^T$ such that $F_i(v_{i,t}) \geq 1 - \frac{1+\varepsilon'}{n}$. Each item independently satisfies this property ($F_i(v_{i,t}) \geq 1 - \frac{1+\varepsilon'}{n}$) with probability $\frac{1+\varepsilon'}{n}$. Hence the probability this does not occur is at most $2\exp\left(-2\varepsilon'^2 T\right)$.

Next, consider $\mathcal{E}_{i2}$ for each agent $i$. The expected contribution of each item to $v_i(A_i^{\text{high}})$ is

$$\mathop{\mathbb{E}}_{Q\sim\text{Unif}[0,1]}\left[F_i^{-1}(Q)\cdot\mathbb{I}\left[Q\geq 1-\frac{1+\varepsilon'}{n}\right]\right]=\frac{1+\varepsilon'}{n}\mathop{\mathbb{E}}_{Q\sim\text{Unif}[1-\frac{1+\varepsilon'}{n},1]}[F_i^{-1}(Q)]$$

$$\leq\frac{1+\varepsilon'}{n}\mathop{\mathbb{E}}_{Q\sim\text{Unif}[1-\frac{1}{n},1]}[F_i^{-1}(Q)].$$

We now use the following multiplicative version of the Chernoff bound,

$$\Pr\left[\sum_i X_i\geq(1+\delta)\sum_i\mathbb{E}[X_i]\right]\leq\exp\left(-\frac{\delta^2}{3}\sum_i\mathbb{E}[X_i]\right),$$

to conclude that the probability that $v_i(A_i^{\text{high}})$ exceeds $T\cdot\frac{(1+\varepsilon')^2}{n}\mathbb{E}_{Q\sim\text{Unif}[1-1/n,1]}[F^{-1}(Q)]\geq(1+\varepsilon')\cdot\mathbb{E}[v_i(A_i^{\text{high}})]$ is at most $\exp\left(-\frac{\varepsilon'^2(1+\varepsilon')\mathbb{E}_{Q\sim\text{Unif}[1-\frac{1}{n},1]}[F_i^{-1}(Q)]}{3n}\cdot T\right)$.

Finally, consider $\mathcal{E}_{i3}$ for each agent $i$. We will show that the expected contribution of each item to $v_i(A_i)$ is at least $\frac{1}{en}\cdot\mathbb{E}_{Q\sim\text{Unif}[1-\frac{1}{n},1]}[F_i^{-1}(Q)]$. Indeed, consider an item such that the quantile for agent $i$ is $Q_i>1-1/n$ while $Q_j<1-1/n$ for all agents $j\neq i$. This occurs with probability $\frac{1}{n}\cdot\left(1-\frac{1}{n}\right)^{n-1}\geq\frac{1}{en}$, and when this occurs, since the algorithm satisfies $\mathcal{P}^*$, it must allocate the item to $i$. Further, when this does occur, the expected value of such an item is $\mathbb{E}_{Q\sim\text{Unif}[1-\frac{1}{n},1]}[F_i^{-1}(Q)]$, since it is independent of the other agent's values. Hence the expectation is at least $\frac{1}{en}\mathbb{E}_{Q\sim\text{Unif}[1-\frac{1}{n},1]}[F_i^{-1}(Q)]$. Finally, we again use a multiplicative Chernoff bound to show that

$$\Pr\left[v_i(A_i)\leq(1-\epsilon')\cdot\frac{T}{en}\mathop{\mathbb{E}}_{Q\sim\text{Unif}[1-\frac{1}{n},1]}[F_i^{-1}(Q)]\right]\leq\exp\left(-\frac{\varepsilon'^2\,\mathbb{E}_{Q\sim\text{Unif}[1-\frac{1}{n},1]}[F_i^{-1}(Q)]}{2en}\cdot T\right).$$

Now, suppose that $\mathcal{E}_{ij}$ hold for all $i\in\mathcal{N}$ and $j\in\{1,2,3\}$. We show that this implies the allocation $A_1,\dots,A_n$ is $(1/e-\varepsilon)$-PO. Fix an arbitrary allocation $A_1',\dots,A_n'$. We show there exists an agent $i\in\mathcal{N}$ such that $v_i(A_i')<\frac{v_i(A_i)}{1/e-\varepsilon}$. First, there must be some agent $i$ such that $|A_i'|\leq T/n$. Since $A_i'$ can be at most as valuable as the most-valuable $T/n$ items, we have

$$v_i(A_i')\leq v_i(A_i^{\text{top}})$$
$$\leq^{(\mathcal{E}_{i1})}v_i(A_i^{\text{high}})$$
$$\leq^{(\mathcal{E}_{i2})}T\cdot\frac{(1+\varepsilon')^2}{n}\mathop{\mathbb{E}}_{Q\sim\text{Unif}[1-1/n,1]}[F^{-1}(Q)]$$
$$\leq^{(\mathcal{E}_{i3})}\cdot\frac{(1+\varepsilon')^2}{(1-\varepsilon')(1/e)}v_i(A_i)$$
$$<\frac{1}{1/e-\varepsilon}v_i(A_i),$$

as needed. $\qquad\square$

# B Proofs missing from Section 4

## B.1 Proof of Theorem 4

Fix an arbitrary, continuous value distribution $D$ and an algorithm $\mathcal{A}$.

As the agents are a priori identical, we can assume without loss of generality that $\mathcal{A}$ gives the first item to agent 1. We will show that, with a positive probability, this decision becomes an irrevocable "mistake," in the sense that agent 2 really liked the item and agent 1 did not. This mistake will make envy-freeness and one-swap PO incompatible.

First, we find values to make this mistake sufficiently bad. Let $g:[0,1]\to[0,1]$ be the function $g(q)=\mathbb{E}[X\mid X\leq F^{-1}(q)]/\mathbb{E}[X]$, which maps a quantile $q$ to the ratio of the expected value

of an item below quantile $q$ to the expected value of an arbitrary item. $g$ is a continuous increasing function with $g(1) = 1$, so there is some quantile $\hat{q} < 1$ such that $g(\hat{q}) \geq 0.9$. Let $q_2^* = \max(\hat{q}, 0.9)$. Since $g$ is increasing, $g(q_2^*) \geq g(\hat{q}) \geq 0.9$. Let $q_1^* = 0.1$, $v_1^* = F^{-1}(q_1^*)$ and $v_2^* = F^{-1}(q_2^*)$. Let $\mathcal{E}^{\text{mistake}}$ be the event that $X_{1,1} < v_1^*$ and $X_{2,1} > v_2^*$. Define $c := \Pr[\mathcal{E}^{\text{mistake}}] = (1 - q_2^*) \cdot q_1^*$ to be the probability that $\mathcal{E}^{\text{mistake}}$ occurs. $D$ is continuous, so $c > 0$. Our lower bound on the probability that the allocation at step $t$ violates either envy-freeness or one-swap PO will only depend on $c$.

Let $\mathcal{E}_j$ be the event that for item $j$ we have that both $X_{1,j} \geq v_1^*$ and $X_{2,j} \leq v_2^*$. If $\mathcal{E}^{\text{mistake}}$ occurs, the only way to maintain one-swap Pareto efficiency is to allocate item $j$ to agent 1 every time $\mathcal{E}_j$ occurs; otherwise, swapping items 1 and $j$ between the two agents yields a Pareto improvement. This constraint will make envy-freeness unlikely.

Let $\mathcal{E}^{\text{manyhigh}}(t)$ be the event $\sum_{j=2}^{t} X_{2,j} \cdot \mathbb{I}[\mathcal{E}_j] \geq (t-1) \cdot 0.7 \cdot \mathbb{E}[X]$. In other words, $\mathcal{E}^{\text{manyhigh}}(t)$ occurs when agent 2 has a high value for items $j$, $2 \leq j \leq t$, for which $\mathcal{E}_j$ occurs (i.e., the items that must be given to agent 1 in order to satisfy one-swap PO). Let $\mathcal{E}^{\text{normalval}}(t)$ denote the event that $\sum_{j=2}^{t} X_{2,j} \leq (t-1) \cdot 1.1 \cdot \mathbb{E}[X]$. We first show that for sufficiently large $t$, the probability that both $\mathcal{E}^{\text{manyhigh}}(t)$ and $\mathcal{E}^{\text{normalval}}(t)$ occur is at least $1/2$. To do so, we prove each event occurs with probability at least $3/4$, and then apply a union bound.

First, since each $X_{1,j}$ and $X_{2,j}$ are independent, $\Pr[\mathcal{E}_j] \geq 0.9 \cdot 0.9 = 0.81$, and $\mathbb{E}[X_{2,j}|\mathcal{E}_j] = \mathbb{E}[X_{2,j} \mid X_{2,j} \leq v_2^*]$. Also, from the definition of $g(\hat{q})$ and the choice of $q_2^*$, $\mathbb{E}[X_{2,j} \mid X_{2,j} \leq v_2^*] \geq 0.9 \cdot \mathbb{E}[X]$. It follows that $\mathbb{E}[X_{2,j} \cdot \mathbb{I}[\mathcal{E}_j]] = \mathbb{E}[X_{2,j}|\mathcal{E}_j] \cdot \Pr[\mathcal{E}_j] \geq 0.729 \cdot \mathbb{E}[X]$. A straightforward Chernoff bound establishes that $\Pr[\mathcal{E}^{\text{manyhigh}}(t)] \geq 3/4$ for $t$ at least $\frac{6}{\mathbb{E}[X]}$.

Let $Y_j = X_{2,j} \cdot \mathbb{I}[\mathcal{E}_j]$ for all $j$. Then, $\mathbb{E}[Y_j] \geq 0.729 \cdot \mathbb{E}[X]$, and $\mathbb{E}[\sum_{j=2}^{T} Y_j] \geq (t-1) \cdot 0.729 \cdot \mathbb{E}[X]$. We are interested in the probability that $\sum_{j=2}^{t} Y_j$ is at least $(t-1) \cdot 0.7 \cdot \mathbb{E}[X]$, i.e., the probability that $\sum_{j=2}^{t} Y_j$ is at least $\frac{0.7}{0.729}$ its expectation.

We use the following Chernoff bound: Let $Y_1, \ldots, Y_n$ be independent random variables that take values in $[0, 1]$, and let $Y$ be their sum. Then, for all $\delta \in [0, 1)$, $\Pr[Y \leq (1 - \delta)\mathbb{E}[Y]] \leq e^{-\frac{\mathbb{E}[Y]\delta^2}{2}}$.

Continuing our derivation:

$$
\begin{aligned}
\Pr\left[\sum_{j=2}^{t} Y_j \geq (t-1) \cdot 0.7 \cdot \mathbb{E}[X]\right] &= \Pr\left[\sum_{j=2}^{t} Y_j \geq \frac{0.7}{0.79}\mathbb{E}[\sum_{j=2}^{t} Y_j]\right] \\
&= 1 - \Pr\left[\sum_{j=2}^{t} Y_j < \frac{0.7}{0.79}\mathbb{E}[\sum_{j=2}^{t} Y_j]\right] \\
&\geq 1 - \Pr\left[\sum_{j=2}^{t} Y_j \leq 0.89\,\mathbb{E}[\sum_{j=2}^{t} Y_j]\right] \\
&\geq 1 - \exp\left(-\frac{\mathbb{E}[\sum_{j=2}^{t} Y_j](0.89)^2}{2}\right),
\end{aligned}
$$

which is at least $3/4$ when $\frac{\mathbb{E}[\sum_{j=2}^{t} Y_j](0.89)^2}{2}$ is at least $\ln(4)$, or, equivalently, if $t \geq 1 + \frac{2\ln(4)}{0.7 \cdot (0.89)^2 \cdot \mathbb{E}[X]}$. Since $\frac{2\ln(4)}{0.7 \cdot (0.89)^2} < 5$ and $\mathbb{E}[X] < 1$, so $t \geq \frac{6}{\mathbb{E}[X]}$ suffices. $\Pr[\mathcal{E}^{\text{normalval}}(t)] \geq 3/4$ follows similarly.

Next, observe that $\mathcal{E}^{\text{manyhigh}}(t) \cap \mathcal{E}^{\text{normalval}}(t)$ is independent of $\mathcal{E}^{\text{mistake}}$, since the two events depend on disjoint sets of independent random variables. Therefore, $\Pr[\mathcal{E}^{\text{mistake}} \cap \mathcal{E}^{\text{manyhigh}}(t) \cap \mathcal{E}^{\text{normalval}}(t)] = \Pr[\mathcal{E}^{\text{mistake}}] \cdot \Pr[\mathcal{E}^{\text{manyhigh}}(t) \cap \mathcal{E}^{\text{normalval}}(t)] \geq c \cdot 1/2$ for $t \geq 6/\mathbb{E}[X]$.

Let $\mathcal{E}_{\text{SPO}}(t)$ and $\mathcal{E}_{\text{EF}}(t)$ be the events that the allocation at step $t$ is one-swap PO, and envy-free, respectively. When $\mathcal{E}^{\text{mistake}} \cap \mathcal{E}^{\text{manyhigh}}(t) \cap \mathcal{E}^{\text{normalval}}(t)$ occur, the allocation cannot be both one-swap PO and envy-free, i.e. $\Pr\left[\overline{\mathcal{E}_{\text{SPO}}(t) \cap \mathcal{E}_{\text{EF}}(t)} \mid \mathcal{E}^{\text{mistake}} \cap \mathcal{E}^{\text{manyhigh}}(t) \cap \mathcal{E}^{\text{normalval}}(t)\right] = 1$. To see this, notice that first, due to $\mathcal{E}^{\text{mistake}}$, the only way to remain one-swap PO is to give each item $j$ to agent

1 every time $\mathcal{E}_j$ occurs. Second, $\mathcal{E}^{\text{manyhigh}}(t)$ ensures that agent 2's value for these items, and hence agent 2's value for agent 1's bundle, is at least $0.7 \cdot (t-1) \cdot \mathbb{E}[X] + v_{2,1}$. Third, $\mathcal{E}^{\text{normalval}}(t)$ ensures that agent 2's value for all items is at most $1.1 \cdot (t-1) \cdot \mathbb{E}[X] + v_{2,1}$, which is strictly less than twice her value for agent 1's bundle. We conclude that the allocation at step $t$ cannot be proportional, and is hence not envy-free. Overall, we have that

$$
\begin{aligned}
\Pr\left[\overline{\mathcal{E}_{\text{SPO}}(t)}\right] + \Pr\left[\overline{\mathcal{E}_{\text{EF}}(t)}\right] &\geq \Pr\left[\overline{\mathcal{E}_{\text{SPO}}(t)} \cup \overline{\mathcal{E}_{\text{EF}}(t)}\right] \\
&= \Pr\left[\overline{\mathcal{E}_{\text{SPO}}(t) \cap \mathcal{E}_{\text{EF}}(t)}\right] \\
&\geq \Pr\left[\overline{\mathcal{E}_{\text{SPO}}(t) \cap \mathcal{E}_{\text{EF}}(t)} \cap \mathcal{E}^{\text{mistake}} \cap \mathcal{E}^{\text{manyhigh}}(t) \cap \mathcal{E}^{\text{normalval}}(t)\right] \\
&= \Pr\left[\overline{\mathcal{E}_{\text{SPO}}(t) \cap \mathcal{E}_{\text{EF}}(t)} \mid \mathcal{E}^{\text{mistake}} \cap \mathcal{E}^{\text{manyhigh}}(t) \cap \mathcal{E}^{\text{normalval}}(t)\right] \cdot \\
&\qquad \cdot \Pr\left[\mathcal{E}^{\text{mistake}} \cap \mathcal{E}^{\text{manyhigh}}(t) \cap \mathcal{E}^{\text{normalval}}(t)\right] \\
&\geq c/2.
\end{aligned}
$$

Therefore, for $t \geq 6/\mathbb{E}[X]$, at least one of $\Pr\left[\overline{\mathcal{E}_{\text{SPO}}(t)}\right]$ and $\Pr\left[\overline{\mathcal{E}_{\text{EF}}(t)}\right]$ is at least $c/4$. We conclude that no algorithm can be both envy-free and one-swap PO with high probability. $\qquad\square$

### B.2 Proof of Theorem 5

Fix a distribution $D$ with CDF $F$ and let $X$ be a random variable with distribution $D$. Fix some $\varepsilon$ to be $(1-\varepsilon)$-welfare-maximizing. Let $\mathcal{E}_1^T$ be the event that the maximum social welfare at time $T$ is at least $1/2 \cdot \mathbb{E}[X] \cdot T$, let $\mathcal{E}_2^T$ be the event that quantile maximization is $c$-strongly-EF for $c = \frac{(\mathbb{E}[X \mid F(X) \geq 1/2] - \mathbb{E}[X])}{4n}$, and let $\mathcal{E}_3^T$ be the event that Algorithm 1 differs from quantile maximization on at most $f(T)$ items from Lemma 7. We first claim that $\mathcal{E}_1^T \cap \mathcal{E}_2^T \cap \mathcal{E}_3^T$ occurs with high probability in $T$. Note that Lemmas 1 and 7 tell us $\mathcal{E}_2^T$ and $\mathcal{E}_3^T$ each occur with high probability, respectively. For $\mathcal{E}_1^T$, the maximum value for each item is in expectation at least the expected value for a single agent $\mathbb{E}[X]$. Hence, a Chernoff bound tells us $\mathcal{E}_1^T$ occurs with probability at least $1 - \exp\left(\frac{-\mathbb{E}[X]T}{8}\right)$, i.e., with high probability. The claim holds because the intersection of a finite number of high probability events occurs with high probability.

Next, note that for sufficiently large $T$, since $f(T) \in o(T)$, $f(T) \leq \frac{(\mathbb{E}[X \mid F(X) \geq 1/2] - \mathbb{E}[X])}{8n} \cdot T$ and $f(T) \leq \varepsilon/2 \cdot \mathbb{E}[X] \cdot T$ (for any fixed $\varepsilon$ that does not depend on $T$). Fix such a sufficiently large $T$. We show that, conditioned on $\mathcal{E}_1^T \cap \mathcal{E}_2^T \cap \mathcal{E}_3^T$, both EF and $(1-\varepsilon)$-welfare hold. Let $A^{QM} = (A_1^{QM}, \ldots, A_n^{QM})$ be the allocation of quantile maximization and $A = (A_1, \ldots, A_n)$ be the allocation of Algorithm 1. Beginning with envy-freeness, we have that for all pairs of agents $i$ and $j$,

$$
\begin{aligned}
v_i(A_i) &\geq^{(\mathcal{E}_3^T)} v_i(A_i^{QM}) - f(T) \\
&\geq^{(\mathcal{E}_2^T)} v_i(A_j^{QM}) - f(T) + \frac{(\mathbb{E}[X \mid F(X) \geq 1/2] - \mathbb{E}[X])T}{4n} \\
&\geq^{(\mathcal{E}_3^T)} v_i(A_j) - 2f(T) + \frac{(\mathbb{E}[X \mid F(X) \geq 1/2] - \mathbb{E}[X])T}{4n} \\
&\geq v_i(A_j),
\end{aligned}
$$

so the allocation is envy-free. Further, noting that $\mathsf{sw}(A^{QM})$ is the maximum social welfare, we have the welfare approximation is at least

$$
\begin{aligned}
\frac{\mathsf{sw}(A)}{\mathsf{sw}(A^{QM})} &= \frac{\mathsf{sw}(A^{QM}) - (\mathsf{sw}(A^{QM}) - \mathsf{sw}(A))}{\mathsf{sw}(A^{QM})} \\
&\geq^{(\mathcal{E}_3^T)} \frac{\mathsf{sw}(A^{QM}) - f(T)}{\mathsf{sw}(A^{QM})} \\
&= 1 - \frac{f(T)}{\mathsf{sw}(A^{QM})}
\end{aligned}
$$

$$\geq^{(\mathcal{E}_1^T)} 1 - \frac{f(T)}{1/2 \cdot \mathbb{E}[X] \cdot T}$$

$$\geq^{(\mathcal{E}_3^T)} 1 - \frac{\varepsilon/2 \cdot \mathbb{E}[X] \cdot T}{1/2 \cdot \mathbb{E}[X] \cdot T}$$

$$= 1 - \varepsilon,$$

as needed. $\qquad\square$

## C  Missing Proofs from Section 5

### C.1  Proof of Theorem 8

We prove that this negative result holds even for an even stronger class of algorithms in which, at each step $t$, the algorithm *selects* quantile thresholds $q_1^t, \ldots, q_n^t \in [0, 1]$ for each agent, and once an item arrives the algorithm observes, for each agent, whether the quantile of their sampled value $Q_{i,t}$ is above or below the threshold $q_i^t$. Note that this provides at least as much information about the fresh item as comparing it to any single prior item, since there is some uncertainty about the values and quantiles of all prior items.

We first focus on the algorithm for a single time-step and show there is a distribution of values such that, regardless of the quantile thresholds selected and allocations made, it cannot do well.

Fix a number of agents $n$ and assume $n \geq 3$. We handle the special case of $n = 2$ at the end of this proof, as it requires a different distribution. For simplicity we consider a distribution that takes values larger than 1; re-scaling (specifically, dividing all values by $2 + \varepsilon$) gives a distribution upped bounded by 1 and does not affect any of our arguments. Consider the value distribution $X$, with

$$X \sim \begin{cases} \text{Unif}[0, \varepsilon] & \text{with probability } 1 - \frac{1}{n}, \\ \text{Unif}[1, 1 + \varepsilon] & \text{with probability } \frac{2}{3n}, \text{ and} \\ \text{Unif}[2, 2 + \varepsilon] & \text{with probability } \frac{1}{3n} \end{cases}$$

for some small $\varepsilon > 0$ to be fixed later. Intuitively, $X$ is a continuous version of a discrete distribution which takes low value (near 0) with probability $1 - \frac{1}{n}$, medium value (near 1) with probability $\frac{2}{3n}$, and high value (near 2) with probability $\frac{1}{3n}$. Let $F_X$ be its CDF. Trivially, the maximum social welfare of $T$ items when all agents have this value distribution is at most $T \cdot (2 + \varepsilon)$.

We show that regardless of what quantile thresholds the algorithm chooses at step $t$ and which decision it makes given the resulting signals, the expected value of the agent receiving item $t$ is at least $(1 - \varepsilon) \cdot \frac{1}{144e}$ away from optimal. To that end, fix arbitrary thresholds $q_1, \ldots, q_n$. First, we partition the agents depending on whether their quantile $q_i$ is above or below $1 - \frac{2n}{3}$. We let $N^{\text{below}} = \{ i \in [n] \mid q_i < 1 - \frac{2n}{3} \}$ and $N^{\text{above}} = \{ i \in [n] \mid q_i \geq 1 - \frac{2n}{3} \}$. Either $|N^{\text{below}}| \geq \lceil n/2 \rceil$ or $|N^{\text{above}}| \geq \lceil n/2 \rceil$; we analyse each case separately. Since $n \geq 3$, we have $\lceil n/2 \rceil \geq 2$.

*Case I: $|N^{below}| \geq \lceil n/2 \rceil$.* In this case, it will be difficult for the algorithm to distinguish between agents in $N^{\text{below}}$ with medium value and those with high value. Consider the event $\mathcal{E}$ that one agent $i^{\max} \in N^{\text{below}}$ has quantile $Q_{i^{\max}} > 1 - \frac{1}{3n}$, one agent $i^{\text{smax}} \in N^{\text{below}}$ has quantile $Q_{i^{\text{smax}}} \in (1 - \frac{2}{3n}, 1 - \frac{1}{3n})$, and all other agents $i \in \mathcal{N} \setminus \{ i^{\max}, i^{\text{smax}} \}$ have quantile $Q_i < 1 - \frac{1}{n}$. First, we show that $\Pr[\mathcal{E}] \geq \frac{1}{72e}$, a constant. To compute this probability, note that there are at least $\lceil n/2 \rceil \cdot (\lceil n/2 \rceil - 1)$ choices of $i^{\max}$ and $i^{\text{smax}}$. Once these have been selected, the probability of $\mathcal{E}$ occurring for this pair of agents is

$$\frac{1}{3n} \cdot \frac{1}{3n} \cdot \left(1 - \frac{1}{n}\right)^{n-2} \geq^{(n \geq 3)} \frac{1}{9n^2} \left(1 - \frac{1}{n}\right)^{n-1} \geq \frac{1}{9en^2}.$$

Since $\lceil n/2 \rceil \cdot (\lceil n/2 \rceil - 1) \geq n^2/8$, we can that conclude $\Pr[\mathcal{E}] \geq \frac{1}{72e}$. Conditioned on $\mathcal{E}$ occurring, $i^{\max}$ has high value, $i^{\text{smax}}$ has medium value, and all other agents have low value. However, from the perspective of the algorithm, two agents ($i^{\max}$ and $i^{\text{smax}}$) give a high signal, and it's equally likely that each of them is the agent with the high value (note that we condition on $\mathcal{E}$). The algorithm must therefore allocate the item to an agent with at most medium value (upper bounded by $1 + \varepsilon$) with probability at least $1/2$, even though an agent with value at least 2 exists. Hence, in this timestep, the

algorithm has an additive error (compared to the optimum welfare) of at least $(1-\varepsilon)$ with probability at least $\frac{1}{144e}$.

*Case II:* $|N^{above}| \geq \lceil n/2 \rceil$. In this case, it will be difficult for the algorithm to distinguish between agents in $N^{above}$ that have medium value and those with low value. Consider the event $\mathcal{E}$ that one agent $i^{max} \in N^{above}$ has quantile $Q_{i^{max}} \in (1 - \frac{1}{n}, 1 - \frac{2}{3n})$ and all other agents $i \in \mathcal{N} \setminus \{i^{max}\}$ have quantile $Q_i < 1 - \frac{1}{n}$. First, we show that $\Pr[\mathcal{E}] \geq \frac{1}{6e}$. Indeed, there are at least $n/2$ choices for $i^{max}$. For a fixed choice of $i^{max}$, the probability of $\mathcal{E}$ occurring is $\frac{1}{3n} \cdot \left(1 - \frac{1}{n}\right)^{n-1} \geq \frac{1}{3en}$, and there are at least $n/2$ choices for $i^{max}$, so $\Pr[\mathcal{E}] \geq \frac{1}{6e}$. Agent $i^{max}$ and the other members of $N^{above}$ (there is at least one more) are indistinguishable to the algorithm as they all have a low signal, so the algorithm must give it to an agent with value at most $\varepsilon$ with probability at least $1/2$ even though an agent with value at least $1$ exists. Hence, in this timestep, the algorithm has an additive error (compared to the optimum welfare) of at least $(1 - \varepsilon)$ with probability at least $\frac{1}{12e}$.

In either case, for every time step, the algorithm has an additive error of at least $(1 - \varepsilon)$ with probability at least $\frac{1}{144e}$, irrespective of the past allocations. As time steps are independent, standard tail bounds give that, for sufficiently small $\varepsilon > 0$, the error is at least $\frac{1-\varepsilon}{1000}T$ with high probability. The optimal social welfare is at most $(2 + \varepsilon) \cdot T$; we conclude the algorithm can be no more than an $0.999-$approximation to welfare.

**The case of two agents.** Finally, we handle the case of two agents. Assume values are drawn from a Unif$[0, 1]$ distribution. Let $q_1, q_2$ be the quantile thresholds selected by the algorithm and, without loss of generality, suppose that $0 \leq q_1 \leq q_2 \leq 1$. At least one of the differences $q_1 - 0, q_2 - q_1, 1 - q_2$ must be at least $1/3$. Suppose $q_2 - q_1 \geq 1/3$ (the other cases are symmetric). We investigate the event that both agents have $Q_i \in [q_1, q_2]$, so that agent 1 signals high and agent 2 signals low, which occurs with probability at least $1/9$. Conditioned on this event, the signals do not provide any additional information, so the algorithm chooses the agent with smaller value at least half of the time. In this case, the expected difference between the larger and smaller values is $1/9$. Hence, the expected difference of the value from the algorithm versus the maximum social welfare is at least $\frac{1}{9} \cdot \frac{1}{2} \cdot \frac{1}{9} = 1/162$ on each item. The maximum social welfare is at most $T$, and we expect the difference to be at least $T/1000$ due to concentration, so the algorithm cannot guarantee more than a .999 approximation, as needed. $\qquad\square$

## C.2 Proof of Lemma 10

Fix such an $\varepsilon, \delta, \tau$, and $\ell$. We claim that a sufficient condition for $\varepsilon$-accuracy is that all agents accept an item with quantile within $q^* \pm \varepsilon/(2n)$. Indeed, note that any sampled quantile outside this range will be classified (as high vs low) correctly. With such an error tolerance, the probability a specific agent's quantile (for a fresh item) falls within this range is at most $\varepsilon/n$. Via a union bound over all $n$ agents, the probability that *no* agent has a quantile (for a fresh item) within this range is at least $1 - \varepsilon$. Hence, all that needs to be shown is that with probability $1 - \delta$, all agents accept an item and the accepted item has quantile within the allowed range.

Since there are $\tau$ trials, there are at most $n\tau$ items tested across all agents. We show that $\ell$ is large enough such that with probability $1 - \delta/2$, all these tests are within $\pm\varepsilon/(6n)$ of the true value. Using Hoeffding's inequality, the probability any specific test fails is at most

$$2\exp\left(-2\left(\frac{\varepsilon}{6n}\right)^2 \cdot \ell\right) = 2\exp\left(-\frac{\varepsilon^2}{18n^2} \cdot \ell\right) \leq 2\exp\left(-\ln\left(\frac{4\tau n}{\delta}\right)\right) = \frac{\delta}{2\tau n},$$

a union bound over all $n\tau$ tests yields the required probability.

Note that under the condition that all the tests are this accurate, since the threshold for acceptance is $\pm\varepsilon/(3n)$, any accepted item will be within $\pm\varepsilon/(2n)$ of $q^*$, as needed. What remains to be shown is that each agent will, with reasonable probability, accept an item. To that end, we need to show that with probability $1 - \delta/2$, all agents will test an item that is within $\pm\varepsilon/(6n)$ of $q^*$. If such an item is tested and the test is accurate, the empirical estimate of its quantile is within $\pm\varepsilon/(3n)$, and the item would hence be accepted. A union bound will then tell us that both of these events would occur with probability $1 - \delta$.

Towards proving that each agent will test an item within $\pm\varepsilon/(6n)$ of $q^*$ with probability $1 - \delta/2$, we use a union bound, showing that each agent individually will *not* sample such an item with

probability at most $\delta/(2n)$. In each of $\tau$ trials, the probability such an item is sampled is $\varepsilon/(3n)$. Hence, the probability no such item is sampled is $\left(1 - \frac{\varepsilon}{3n}\right)^\tau$. We then have that

$$
\begin{aligned}
\left(1 - \frac{\varepsilon}{3n}\right)^\tau = \left(1 - \frac{1}{\frac{3n}{\varepsilon}}\right)^\tau & \\
&= \left(\left(1 - \frac{1}{\frac{3n}{\varepsilon}}\right)^{\frac{3n}{\varepsilon}}\right)^{\tau \cdot \frac{\varepsilon}{3n}} \\
&\le \left(e^{-1}\right)^{\tau \cdot \frac{\varepsilon}{3n}} \qquad\qquad (1 - 1/x)^x \le e^{-1} \text{ for all } x \ge 1 \\
&= e^{-\tau \cdot \frac{\varepsilon}{3n}} \\
&\le e^{-\ln(2n/\delta)} \\
&= \frac{\delta}{2n},
\end{aligned}
$$

as needed. $\qquad\qquad\qquad\qquad\qquad\qquad\qquad\qquad\qquad\qquad\qquad\qquad\qquad\qquad\qquad\qquad$ $\square$

### C.3 Proof of Lemma 11

First, we prove that for all $k \ge 10n$, epoch $k$ is $\varepsilon_k$-accurate with probability $\delta_k$ for $\varepsilon_k = 3n/k^2$ and $\delta_k = 2ne^{-k}$. Since $k > 3n$ these are valid values between 0 and 1. Hence, we simply need to check that the $\tau$ and $\ell$ inequalities hold for the number of trials and number of test items specified in Algorithm 2. For arbitrary epoch $k$,

$$
\frac{\ln(2n/\delta_k)}{\varepsilon_k/(3n)} = \ln\left(e^k\right)k^2 = k^3,
$$

so the number of trials is sufficiently large. Further,

$$
\begin{aligned}
\frac{18n^2}{\varepsilon_k^2} \ln\left(\frac{4k^3 n}{\delta_k}\right) &= 2k^4 \cdot \ln\left(2k^3 e^k\right) \\
&\le 2k^4 \cdot \ln\left(k^4 e^k\right) & (k \ge 2) \\
&= 2k^4 \cdot (k + 4\ln k) \\
&\le 2k^4 \cdot (k + 4k) & (\ln k < k) \\
&\le 10k^5 \\
&\le k^6. & (k \ge 10)
\end{aligned}
$$

Recall that $k(t)$ is defined as the epoch of item $t$. As in the proof of Lemma 7, we characterize deviations from the ideal algorithm in four ways.

1. Item $t$ was allocated in one of the first $10n - 1$ epochs; that is, $k(t) < 10n$.
2. Item $t$ was allocated during the sampling phase of epoch $k(t) \ge 10n$.
3. Item $t$ was allocated during the ranking phase of epoch $k(t) \ge 10n$, which was $\varepsilon_{k(t)}$-accurate.
4. Item $t$ was allocated during the ranking phase of epoch $k(t) \ge 10n$, which was not $\varepsilon_{k(t)}$-accurate.

We say an item $t$ is incorrect (incorrectly allocated) when it is given to an agent with non-maximum quantile for it. We show that the number of mistakes in each category are bounded by $10^{20}n^{19}$, $2T^{5/9}$, $7nT^{17/18}$ and $1.3 \cdot 10^{16}n$ respectively, with high probability. This implies, via a union bound, that the total number of mistakes is at most the sum of these quantities, or $O(\text{poly}(n) \cdot T^{17/18})$, with high probability.

The number of items in category 1, is at most

$$
\sum_{k=1}^{10n} k^9 + k^{18} \le (10n)^{10} + (10n)^{19} \le 10^{20}n^{19}
$$

Notice that $T \geq \sum_{k=1}^{k(T)-1} k^9 + k^{18} \geq (k(T) - 1)^{18}$, and therefore $k(T) \leq 2T^{1/18}$.

For the second category, since $k(T) \leq 2T^{1/18}$, the total number of items in the sampling phase is (with probability 1) upper bounded by

$$\sum_{k=1}^{k(T)} k^9 \leq k(T)^{10} \leq 2T^{5/9}.$$

For the third category, note that each item $t$ in this category has probability $\varepsilon_{k(t)}$ of being incorrect. The expected number of mistakes is at most

$$\sum_{k=10n}^{k(T)} \varepsilon_{k(t)} k^{18} = \sum_{k=10n}^{k(T)} 3nk^{16} \leq 3nk(T)^{17} \leq 6nT^{17/18}.$$

Using Hoeffding's inequality we get that with high probability the number of mistakes is at most $7nT^{17/18}$, since a deviation of $nT^{17/18}$ occurs with probability at most $\exp(-2n^2T^{17/9}/T) = \exp(-2n^2T^{8/9})$.

For the fourth category, the expected number of items in this category is at most

$$\sum_{k=10n}^{k(T)} \delta_k k^{18} = 2n \sum_{k=10}^{k(T)} \frac{k^{18}}{e^k} \leq 2n \sum_{k=1}^{\infty} \frac{k^{18}}{e^k} \leq 1.3 \cdot 10^{16} n.$$

Using Markov's inequality we have that the number of mistakes is at most $1.3 \cdot 10^{16} n \ln(T)$ with probability at least $1 - \ln(T)$, i.e., with high probability. $\square$

### C.4 Proof of Theorem 9

The proof is nearly identical to the proof of Theorem 5. Fix a distribution $D$ with CDF $F$ and let $X$ be a random variable with distribution $D$. Fix some $\varepsilon$ to be $(1 - 1/e) - \varepsilon$ welfare maximizing. Let $\mathcal{E}_1^T$ be the event that the maximum social welfare at time $T$ is at least $1/2 \cdot \mathbb{E}[X] \cdot T$, let $\mathcal{E}_2^T$ be the event the ideal threshold algorithm is $c$-strongly-EF for $c = \frac{(\mathbb{E}[X \mid F(X) \geq 1/2] - \mathbb{E}[X])}{4n}$, let $\mathcal{E}_3^T$ be the event that the ideal threshold algorithm is a $(1 - 1/e)^2 - \varepsilon/2$ approximation to welfare, and let $\mathcal{E}_4^T$ be the event that Algorithm 2 differs from the ideal threshold algorithm on at most $f(T)$ items from Lemma 11. We first claim that $\mathcal{E}_1^T \cap \mathcal{E}_2^T \cap \mathcal{E}_3^T \cap \mathcal{E}_4^T$ occurs with high probability in $T$. Note that Lemmas 2, 1 ,and 11 tell us each of $\mathcal{E}_2^T, \mathcal{E}_3^T$, and $\mathcal{E}_4^T$ occur with high probability. For $\mathcal{E}_1^T$, the maximum value for each item is in expectation at least the expected value for a single agent $\mathbb{E}[X]$. Hence, a Chernoff bound tells us $\mathcal{E}_1^T$ occurs with probability at least $1 - \exp\left(\frac{-\mathbb{E}[X]T}{8}\right)$, i.e., with high probability. The claim holds because the intersection of a finite number of high probability events occurs with high probability.

Next, note that for sufficiently large $T$, since $f(T) \in o(T)$, $f(T) \leq \frac{(\mathbb{E}[X \mid F(X) \geq 1/2] - \mathbb{E}[X])}{8n} \cdot T$ and $f(T) \leq \varepsilon/4 \cdot \mathbb{E}[X] \cdot T$. Fix such a sufficiently large $T$. We show that conditioned on $\mathcal{E}_1^T \cap \mathcal{E}_2^T \cap \mathcal{E}_3^T \cap \mathcal{E}_4^T$, both EF and $((1 - 1/e)^2 - \varepsilon)$-welfare hold. Recall that a "difference" between Algorithm 2 and the ideal threshold algorithm refers to different distributions over the agents that get some item (i.e., a different randomized allocation). In order to make statements about envy-freeness and efficiency we need a way to argue about the differences between the algorithms ex-post. However, notice that without loss of generality we can couple the decision made by the two algorithms when randomized allocation is the same; that is, when Algorithm 2 does not differ from the ideal threshold algorithm we can assume without loss of generality that the agent who gets the item is the same. Let $A^{IT} = (A_1^{IT}, \ldots, A_n^{IT})$ be the allocation of the ideal threshold algorithm and $A = (A_1, \ldots, A_n)$ be the allocation of Algorithm 2. Beginning with envy-freeness, we have that for all pairs of agents $i$ and $j$,

$$v_i(A_i) \geq^{(\mathcal{E}_4^T)} v_i(A_i^{IT}) - f(T)$$
$$\geq^{(\mathcal{E}_2^T)} v_i(A_j^{IT}) - f(T) + \frac{(\mathbb{E}[X \mid F(X) \geq 1/2] - \mathbb{E}[X])T}{4n}$$

$$\geq^{(\mathcal{E}_4^T)} v_i(A_j) - 2f(T) + \frac{(\mathbb{E}[X \mid F(X) \geq 1/2] - \mathbb{E}[X])T}{4n}$$

$$\geq v_i(A_j),$$

so the allocation is envy-free. Let $A^*$ be a welfare-maximizing algorithm. For the welfare approximation, we then have

$$\frac{\mathsf{sw}(A)}{\mathsf{sw}(A^*)} = \frac{\mathsf{sw}(A^{IT}) - (\mathsf{sw}(A^{IT}) - \mathsf{sw}(A))}{\mathsf{sw}(A^*)}$$

$$\geq^{(\mathcal{E}_4^T)} \frac{\mathsf{sw}(A^{IT}) - f(T)}{\mathsf{sw}(A^*)}$$

$$= \frac{\mathsf{sw}(A^{IT})}{\mathsf{sw}(A^*)} - \frac{f(T)}{\mathsf{sw}(A^*)}$$

$$\geq^{(\mathcal{E}_3^T)} (1 - 1/e)^2 - \varepsilon/2 - \frac{f(T)}{\mathsf{sw}(A^*)}$$

$$\geq^{(\mathcal{E}_1^T)} (1 - 1/e)^2 - \varepsilon/2 - \frac{f(T)}{1/2 \cdot \mathbb{E}[X] \cdot T}$$

$$\geq (1 - 1/e)^2 - \varepsilon/2 - \frac{\varepsilon/4 \cdot \mathbb{E}[X] \cdot T}{1/2 \cdot \mathbb{E}[X] \cdot T}$$

$$= (1 - 1/e)^2 - \varepsilon,$$

as needed. $\qquad\square$

# D  Missing Proofs from Section 6

## D.1  Proof of Theorem 12

Suppose for contradiction that there is an algorithm $\mathcal{A}$ so that for all bounded continuous distributions $(X_1, X_2)$ there exists a $T^* = T^*(X_1, X_2)$ where for all $t \geq T^*$, $\mathcal{A}$ is envy-free and $c$-PO with probability $p$ with $p > 2/3$ for some constant $c > \frac{1+\sqrt{5}}{4}$. Hence, there is some $\varepsilon$ such that $p > 2/3 + \varepsilon$ and $1/c < \frac{4}{1+\sqrt{5}} - \varepsilon = \sqrt{5} - 1 - \varepsilon$.

Consider two distributions $D_F$ and $D_S$; we describe these later in the proof. Consider the three instances $I_0 = (D_F, D_F)$, $I_1 = (D_S, D_F)$ and $I_2 = (D_F, D_S)$.

Let $\mathcal{E}_j^{\mathcal{A},t}$ be the event that $\mathcal{A}$ is envy-free and $c$-PO on instance $I_j$ at time $t$ for $j \in \{0, 1, 2\}$. By construction, $\Pr\left[\mathcal{E}_j^{\mathcal{A},t}\right] \geq 2/3 + \varepsilon$ for all $j \in \{0, 1, 2\}$ and $t \geq T^*$.

Let $z$ be a parameter we will fix later in the proof, and let $Z_i^t = \mathbb{I}\{Q_{i,t} \geq 1 - z\}$ for $i = \{1, 2\}$. Observe that $Z_1^t \cdot Z_2^t$ is 1 with probability $z^2$ and 0 otherwise. The following events characterize a specific notion of a "nice" sample, in which the number of items with high quantiles for both agents is near its expectation: $\mathcal{E}_1^T = \mathbb{I}\{|\frac{1}{T}\sum_{t=1}^T Z_1^t \cdot Z_2^t - z^2| < \delta\}$, $\mathcal{E}_2^T = \mathbb{I}\{|\frac{1}{T}\sum_{t=1}^T Z_1^t - z| < \delta\}$, and $\mathcal{E}_3^T = \mathbb{I}\{|\frac{1}{T}\sum_{t=1}^T Z_2^t - z| < \delta\}$ for some $\delta > 0$. By Hoeffding's inequality, $\Pr[\bar{\mathcal{E}}_1^T] = \Pr\left[|\frac{1}{T}\sum_{t=1}^T Z_1^t \cdot Z_2^t - z^2| \geq \delta\right] \leq 2\exp\left(-2T\delta^2\right)$. It follows that for $T \geq \log(2/\varepsilon)/(2\delta^2)$, $\Pr[\bar{\mathcal{E}}_1^T] \leq \varepsilon$. Similarly, for $T \geq \log(2/\varepsilon)/(2\delta^2)$, it holds that $\Pr[\bar{\mathcal{E}}_2^T] \leq \varepsilon$, and $\Pr[\bar{\mathcal{E}}_3^T] \leq \varepsilon$. Consider an arbitrary $T > T_{\max} = \max\{T_0, T_1, T_2, \log(2/\varepsilon)/(2\delta^2)\}$. Applying a union bound,

$$\Pr\left[\bar{\mathcal{E}}_0^{\mathcal{A},T} \cup \bar{\mathcal{E}}_1^{\mathcal{A},T} \cup \bar{\mathcal{E}}_2^{\mathcal{A},T} \cup \bar{\mathcal{E}}_1^T \cup \bar{\mathcal{E}}_2^T \cup \bar{\mathcal{E}}_3^T\right] \leq \sum_{i=0}^2 \Pr\left[\bar{\mathcal{E}}_i^{\mathcal{A},T}\right] + \sum_{i=1}^3 \Pr\left[\bar{\mathcal{E}}_i^T\right] < 3 \cdot \left(\frac{1}{3} - \varepsilon\right) + 3\varepsilon = 1.$$

It follows that $\Pr\left[\mathcal{E}_0^{\mathcal{A},T} \cap \mathcal{E}_1^{\mathcal{A},T} \cap \mathcal{E}_2^{\mathcal{A},T} \cap \mathcal{E}_1^T \cap \mathcal{E}_2^T \cap \mathcal{E}_3^T\right] > 0$. Therefore, there must exist a sequence of $T$ items whose quantiles satisfy all of $\mathcal{E}_1^T$, $\mathcal{E}_2^T$, and $\mathcal{E}_3^T$, and, since $\mathcal{A}$ does not have access to the items' values, there must exist an allocation $A^T$ for these $T$ items (in the support of $\mathcal{A}$) that is EF and $c$-PO, no matter which of $I_0$, $I_1$ or $I_2$ the values were taken from. Let $q^T =$

$\{(q_1(t), q_2(t))\}_{t=1}^T$ be these items' quantiles. Let $H_B = \{t \in [T] : q_1(t) \geq 1-z \text{ and } q_2(t) \geq 1-z\}$ be the items for which $Z_1^t \cdot Z_2^t = 1$, and $H_1 = \{t \in [T] : q_1(t) \geq 1-z\}$ the items for which $Z_1^t = 1$.

Set distributions $D_F = \text{Unif}[1-w, 1]$ and $D_S$, under which each item is $\text{Unif}[0, w]$ with probability $z$ and at $\text{Unif}[1-w, 1]$ with probability $1-z$, for some small positive $w$ that we fix later in the proof.

We have that some agent receives at most half the items in $H_B$; without loss of generality this is agent 2, i.e., $|A_2^T \cap H_B| \geq |H_B|/2$. We show that there exists a feasible more than $1/c$ Pareto-improvement under the values in $I_1$. To that end, we compare $A^T$ to the allocation $\hat{A}$ where $\hat{A}_1 = H_1$ and $\hat{A}_2 = \bar{H}_1$.

We next bound the utilities of each agent under $A^T$ and $\hat{A}$. Beginning with agent 1, we have

$$
\begin{aligned}
u_1(\hat{A}_1) &= u_1(H_1) \\
&\geq |H_1| \cdot (1-w) \\
&\geq^{(\mathcal{E}_1^T)} T \cdot (z - \delta)(1-w) \\
&= T(z - \delta - zw + \delta w) \\
&\geq T(z - \delta - w)
\end{aligned}
$$

and

$$
\begin{aligned}
u_1(A_1) &\leq w \cdot |A_1 \cap \bar{H}_1| + 1 \cdot |A_1 \cap H_1| \\
&\leq T \cdot w + |H_1| - |A_2 \cap H_1| \\
&\leq T \cdot w + |H_1| - |A_2 \cap H_B| \\
&\leq^{(\mathcal{E}_2^T)} T \cdot w + T(z + \delta) - |A_2 \cap H_B| \\
&\leq T \cdot w + T(z + \delta) - |H_B|/2 \\
&\leq^{(\mathcal{E}_1^T)} T \cdot w + T(z + \delta) - T(z^2 - \delta)/2 \\
&= T(z - z^2/2 + w + 3\delta/2).
\end{aligned}
$$

Together, these imply

$$
\frac{u_1(\hat{A}_1)}{u_1(A_1^T)} \geq \frac{z - \delta - w}{z - z^2/2 + w + 3\delta/2} = \frac{2z - 2\delta - 2w}{2z - z^2 + 2w + 3\delta}.
$$

Next, we consider agent 2. We have

$$
\begin{aligned}
u_2(\hat{A}_2) &= u_2(\bar{H}_1) \\
&\geq (1-w)|\bar{H}_1| \\
&= (1-w)(T - |H_1|) \\
&\geq^{(\mathcal{E}_2^T)} (1-w)T \cdot (1 - (z + \delta)) \\
&= T(1 - z - \delta - w + wz + w\delta) \\
&\geq T(1 - z - \delta - w).
\end{aligned}
$$

By $\mathcal{E}_0^{\mathcal{A}}$, $A^T$ is envy-free on $I_0$. It follows that $|A_1^T| \geq (1-w)|A_2^T|$. Since $|A_1^T| + |A_2^T| = T$, we have that $|A_2^T| \leq \frac{1}{2-w}T$. Hence, $u_2(A_2^T) \leq |A_2^T| \leq \frac{1}{2-w}T$. Combining these, we have

$$
\frac{u_2(\hat{A}_2)}{u_2(A_2^T)} = \frac{1 - z - \delta - w}{\frac{1}{2-w}} = 2 - 2z - 2\delta - 2w - w + wz + w\delta + w^2 \geq 2 - 2z - 2\delta - 3w.
$$

Choose $z = \frac{3 - \sqrt{5}}{2}$. Note that $z^2 = \frac{7 - 3\sqrt{5}}{2}$. Choose $\delta, w < \varepsilon/25$. We then have,

$$
\begin{aligned}
\frac{u_1(\hat{A}_1)}{u_1(A_1^T)} &> \frac{3 - \sqrt{5} - \varepsilon/5}{(\sqrt{5} - 1)/2 + \varepsilon/5} \\
&= \frac{3 - \sqrt{5}}{(\sqrt{5} - 1)/2 + \varepsilon/5} - \frac{\varepsilon/5}{(\sqrt{5} - 1)/2 + \varepsilon/5}
\end{aligned}
$$

$$
\begin{aligned}
&> \frac{3 - \sqrt{5}}{(\sqrt{5} - 1)/2 + \varepsilon/5} - \frac{2\varepsilon}{5} && (\tfrac{\sqrt{5}-1}{2} + \tfrac{\varepsilon}{5} > 1/2) \\
&> \frac{3 - \sqrt{5}}{(\sqrt{5} - 1)/2 \cdot (1 + 2\varepsilon/5)} - \frac{2\varepsilon}{5} && (\sqrt{5} - 1 > 1) \\
&= (\sqrt{5} - 1) \cdot \frac{1}{1 + 2\varepsilon/5} - \frac{2\varepsilon}{5} \\
&> (\sqrt{5} - 1) \cdot (1 - 2\varepsilon/5) - \frac{2\varepsilon}{5} \\
&> (\sqrt{5} - 1) - \varepsilon/2 - \frac{2\varepsilon}{5} && ((\sqrt{5} - 1) \cdot 2/5 < 1/2) \\
&> \sqrt{5} - 1 - \varepsilon \\
&> 1/c
\end{aligned}
$$

and

$$
\frac{u_2(\hat{A}_2)}{u_2(A_2^T)} > 2 - (3 - \sqrt{5}) - \varepsilon/5 > \sqrt{5} - 1 - \varepsilon > 1/c,
$$

so this is more than a $1/c$ Pareto Imrovement. □

### D.2   Proof of Theorem 13

The proof of envy-freeness for each algorithm is nearly identical to Theorems 5 and 9 respectively; we show it here for completeness. We focus on Algorithm 1. The proof for Algorithm 2 goes through identically with all occurances of quantile maximization replaced with the ideal threshold algorithm and all occurances of Lemma 7 replaced with Lemma 11.

Fix a distributions $D_1, \ldots, D_n$ with CDFs $F_1, \ldots, F_n$ and let $X_i$ be a random variable with distribution $D_i$. Fix some $\varepsilon$ to be $(1/e - \varepsilon)$-PO. Let $E = \min_{i \in \mathcal{N}} \mathbb{E}[X_i]$ be the minimum expected value for all agents. Let $\mathcal{E}_1^T$ be the event that each agent $i$'s value for their bundle at time $T$ is at least $1/(2n) \cdot E \cdot T$, let $\mathcal{E}_2^T$ be the even that quantile maximization is $c$-strongly-EF for $c = \min_{i \in \mathcal{N}} \frac{(\mathbb{E}[X_i \mid F_i(X_i) \geq 1/2] - \mathbb{E}[X_i])}{4n}$, let $\mathcal{E}_3^T$ be the event that quantile maximization is a $(1/e - \varepsilon/2)$-PO, and let $\mathcal{E}_4^T$ be the event that Algorithm 1 differs from quantile maximization on at most $f(T)$ items from Lemma 7. We first claim that $\mathcal{E}_1^T \cap \mathcal{E}_2^T \cap \mathcal{E}_3^T \cap \mathcal{E}_4^T$ occurs with high probability in $T$. Note that Lemmas 1, 3, and 7 tell us $\mathcal{E}_2^T, \mathcal{E}_3^T$, and $\mathcal{E}_4^T$ each occur with high probability, respectively. For $\mathcal{E}_1^T$, note that under quantile maximization, the probability each agent $i$ receives an item is exactly $1/n$ and the expected value conditioned on receiving the item is at least $\mathbb{E}[X_i] \geq E$. Hence, the expected contribution of each item to $v_i(A_i)$ is at least $1/n \cdot E$. A Chernoff bound then tells us $\mathcal{E}_1^T$ holds for agent $i$ with probability at least $1 - \exp\left(\frac{-ET}{8n}\right)$. A union bound over all agent's tells us this occurs simultaneously for all agents with probability at least $1 - n \exp\left(\frac{-ET}{8n}\right)$, i.e., with high probability. The claim holds because the intersection of a finite number of high probability events occurs with high probability.

Next, note that for sufficiently large $T$, since $f(T) \in o(T)$, $f(T) \leq \min_{i \in \mathcal{N}} \frac{(\mathbb{E}[X_i \mid F_i(X_i) \geq 1/2] - \mathbb{E}[X_i])}{8n} \cdot T$ and $f(T) \leq \varepsilon/(4n) \cdot ET$. Fix such a sufficiently large $T$. We show that conditioned on $\mathcal{E}_1^T \cap \mathcal{E}_2^T \cap \mathcal{E}_3^T \cap \mathcal{E}_4^T$, both EF and $(1/e - \varepsilon)$-PO hold. Let $A^{QM} = (A_1^{QM}, \ldots, A_n^{QM})$ be the allocation of quantile maximization and $A = (A_1, \ldots, A_n)$ be the allocation of Algorithm 2. Beginning with envy-freeness, we have that for all pairs of agents $i$ and $j$,

$$
\begin{aligned}
v_i(A_i) &\geq^{(\mathcal{E}_4^T)} v_i(A_i^{QM}) - f(T) \\
&\geq^{(\mathcal{E}_2^T)} v_i(A_j^{QM}) - f(T) + \min_{i \in \mathcal{N}} \frac{(\mathbb{E}[X_i \mid F_i(X_i) \geq 1/2] - \mathbb{E}[X_i])}{4n} \\
&\geq^{(\mathcal{E}_4^T)} v_i(A_j) - 2f(T) + \min_{i \in \mathcal{N}} \frac{(\mathbb{E}[X_i \mid F_i(X_i) \geq 1/2] - \mathbb{E}[X_i])}{4n} \\
&\geq v_i(A_j),
\end{aligned}
$$

so the allocation is envy-free. Next, we show that for all agents $\frac{v_i(A_i)}{v_i(A_i^{QM})} \geq 1 - \varepsilon/2$. Since $A^{QM}$ is $(1/e - \varepsilon/2)$-PO under $\mathcal{E}_3^T$, this implies that $A$ is a $(1/e - \varepsilon/2)(1 - \varepsilon/2) \geq 1/e - \varepsilon$ approximation to PO as well.

To that end, for each agent $i$ we have

$$
\begin{aligned}
\frac{v_i(A_i)}{v_i(A_i^{QM})} &= \frac{v_i(A_i^{QM}) - (v_i(A_i^{QM}) - v_i(A_i))}{v_i(A_i^{QM})} \\
&= 1 - \frac{(v_i(A_i^{QM}) - v_i(A_i))}{v_i(A_i^{QM})} \\
&\geq^{(\mathcal{E}_4^T)} 1 - \frac{f(T)}{v_i(A_i^{QM})} \\
&\geq^{(\mathcal{E}_1^T)} 1 - \frac{f(T)}{ET/(2n)} \\
&\geq 1 - \frac{\varepsilon \cdot ET/(4n)}{ET/(2n)} \\
&= 1 - \varepsilon/2,
\end{aligned}
$$

as needed. $\qquad\square$