# OpenReview forum: "Dynamic Fair Division with Partial Information"
_NeurIPS.cc/2022/Conference — NeurIPS 2022 Accept_

### Official Review · Reviewer_rhEt · 2022-06-23

**Rating:** 6
**Confidence:** 3
**Soundness:** 3 good
**Presentation:** 4 excellent
**Contribution:** 3 good

**Summary:**

The paper considers notions of fairness in an online (dynamic) input setting, wherein items arrive one at a time and must be irrevocably allocated to one of the $n$ agents of the system. Specifically, the authors focus on the "partial information" setting where the algorithm has access to ordinal, rather than cardinal information. This is an interesting problem context that, in spite of its increased complexity, is shown to allow for strong fairness guarantees. The paper presents asymptotically tight results for envy-freeness and approximation social welfare maximization under different partial information assumptions (with high probability arguments).

**Questions:**

Have the authors considered other input models (such as adversarial or random order)? Is the iid problem setting the only one in which such strong (and asymptotically tight) results be achieved?



**Ethics Review Area:**

["I don’t know"]

**Limitations:**

The authors mention the correlated input model as a future direction (that is not considered in the submission). I think the above mentioned input models should also be considered limitations of the present work as the iid input is necessarily the strongest assumption.

**Strengths And Weaknesses:**

Strengths:
1. The writing is very clear and easy to follow throughout the paper. Discussion of the results and presentation of their proofs is also nicely presented.

2. The problem context is very interesting and seems to have only been considered in a handful of papers. Additionally, all presented results are asymptotically tight (proofs seem correct) which nicely opens and closes certain questions.

Weaknesses:
1. The authors appear to be missing some important citations that discuss related work. In particular, "Dynamic Fair Division with General Valuations" (Li el al 2018) seems to be a notable omission. Other papers of interest that are not cited: "A simple online fair division problem" (Bogomolnaia et al. 2019) and "Online Nash Social Welfare Maximization with Predictions" (Banerjee et al 2022).

2. While the problem context is well-defined, it does feel that the motivation for it is somewhat lacking. A more extensive discussion of the related work and how the present work fits into that context + some discussion and concluding remarks would be beneficial in this regard for the uninitiated reader.

3. Though most of the writing and proofs are clear, it might be beneficial to defer more of the highly analytic steps to the appendix in favor of a more intuitive discussion of the result (which could in turn save some space to include the above).

4. The iid input assumption seems a bit weak. Do any results exist in the partial information model for the adversarial or random order input models?

---

> ### Author Response · Authors · 2022-07-31
> **Response to Reviewer rhEt**
>
> Thank you for your review and constructive feedback. Below we answer your question.
>
> - “Have the authors considered other input models (such as adversarial or random order)? Is the iid problem setting the only one in which such strong (and asymptotically tight) results be achieved?”
>
> In the adversarial model, the answer is known: Even with known values, it is impossible to simultaneously achieve envy-freeness and Pareto efficiency. In fact, “random allocation” is optimal in terms of envy, and everything with smaller envy than “dictatorship” is as inefficient as “random allocation” (see [Zeng and Psomas, 2020]). And, both “random allocation” and “dictatorship” can be executed in the partial information model, giving a clear picture of what can and what cannot be done.

---

### Official Review · Reviewer_c1uw · 2022-07-11

**Rating:** 6
**Confidence:** 4
**Soundness:** 3 good
**Presentation:** 3 good
**Contribution:** 4 excellent

**Summary:**

The paper studies a standard sequential item allocation problem where items need to be irrevocably allocated to agents upon arrival. The goal is to achieve some notion of fairness (e.g. envyfreeness) and efficiency (e.g. Pareto efficiency).  This paper extends the previous works by focusing on achieving these properties only through partial information (e.g. ordinal relative ranking of items). The authors focus on two main models

1)	Unbounded memory model where an algorithm learns the relative ranking of the item compared to prior items, and
2)	Bounded memory model where each agent only “remembers” a single prior item and the algorithm can only compare against that item.

The paper studies agents with i.i.d (known or unknown) distributions as well as non-i.i.d  distributions. In both cases, the authors devise careful sampling algorithms to achieve envyfreeness and some approximation to welfare with high probability.


**Questions:**

Phase transitions: The paper only focuses on achieving envy-freeness at the end in contrast to satisfying some notion of fairness at each step. If we consider fairness at each step (or even epoch), what type of EF guarantees can we expect? Is there going to be a phase transition at some step similar to the phase transition shown by Dickerson et al.? If yes, when would the phase transition occur?


In the model that takes ordinal information as rankings, the preliminaries section does not properly explain how to deal with weak preferences or ties. How does the algorithm deal with ties when eliciting ranks of the fresh item? Can you elaborate on this?

Updating memory: in Algorithm 2, at each epoch an agent’s memory is updated with comparison with (prior) k^6 items. What is the reason behind such updating mechanism? If an agent’s error is not within the tolerance, an arbitrary item is chosen to be remembered. This seems a bit arbitrary (no pun intended). Is the reason mainly due to assumptions on distributions?
Wouldn’t it be more beneficial to remember a certain item (say the one with the average highest ranking)?

Algorithm 1 uses more memory compared to Algorithm 2 but they both achieve the same guarantee on the non-iid model. However, Algorithm 1 offers shorter epochs. Could Algorithm 1’s bound improve with longer epochs? In other words, is the bound tight both for iid and non-iid models?

The crux of the paper is assuming envyfreeness and devising exploration techniques to achieve reasonable welfare approximations. I wonder whether it is possible to improve on efficiency (or its approximations) by relaxing the fairness notion to EF1, proportionality, etc.?
The first question is whether the impossibility results in Theorem 4 or 12 still hold.

In general, simultaneously achieving fairness and strategyproofness is a challenging task; and it is often impossible in fair division. I wonder if there is a way to achieve strategyproofness under the settings of this paper (e.g i.i.d bounded or unbounded models)?
Let me elaborate: in fair division models, strategyproofness is quite challenging to achieve in general (there are several incompatibility results even with relaxations to fairness). However, there may be some hope in the models with i.i.d distributions, or models that limit elicitation process (such as this paper), and/or when randomization can help alleviate the incompatibilities.


Proof of theorem 4 constructs an instance with two agents where an arbitrary agent receives an item that is not liked so much by it, but it is liked by the other agent. Although constructing such examples/events is easy, the argument is quite neat and shows the limitations of satisfying EF and a weak efficiency notion with high probability even in the i.i.d model.


**Limitations:**

No. The authors do not explicitly discuss any limitations to their results. In fact, I encourage the authors to include a concluding remarks section and discuss not only the limitations (or boundaries) of their results but also some potential future directions.

**Strengths And Weaknesses:**

Strengths:

-	Fair division is a fundamental research direction within AI and computational social choice; the dynamic aspects of fair division, whether it is with respect to dynamic item allocation or elicitations are all intriguing directions with practical value. The current paper makes progress in three different dimensions: achieving fairness and efficiency, dynamic resource allocation, and elicitation frameworks and information loss (due to ordinal vs. cardinal elicitation formats).

-	It is well known that when there are more items than agents (larger than a logarithmic factor), envyfree allocations exist with high probability (Dickerson et al. 2014). While achieving envyfreeness or efficiency separately are trivial under distributional assumptions, simultaneously achieving both is a challenge. Thus, the current paper focuses on sampling/exploration frameworks to improve efficiency bounds (while EF is easy) with unbounded memory (and thus, comparisons). The most surprising result is that the authors show that you can achieve these bounds for efficiency with high probability even if agents are forgetful, i.e., they can only compare a new item with a single prior item.

-	The results in the paper including incompatibility results (Theorems 4 and 12) and the proof of the two algorithms involve challenging (and non-trivial) arguments and seem to be sound--although the write up of the proofs can be improved (see my next comments).

Weaknesses:

-	The write up of the paper can become quite challenging at various points. There is a general assumption that a reader should be more or less familiar with the techniques, reasons they work, etc. which can make the paper inaccessible to a general NeurIPS audience. On the other hand, even for a researcher in computational social choice and fair division, some sections and explanations lack sufficient intuition, which could make the paper hard to follow.

-	The proofs are written mainly formally, which is a plus. However, I was not able to fully follow some of the proof arguments, perhaps due to missing steps. It could be nice to include full proofs and provide intuition on the most challenging aspects of the proof (some steps are easy to see).

-	It is a bit awkward to end a paper with no conclusions or mention of future directions. The ending of the paper seems a bit abrupt.

---

> ### Author Response · Authors · 2022-07-31
> **Response to Reviewer c1uw**
>
> Thank you for your review and constructive feedback. Below we answer your questions.
>
> - “If we consider fairness at each step (or even epoch), what type of EF guarantees can we expect? Is there going to be a phase transition at some step similar to the phase transition shown by Dickerson et al.? If yes, when would the phase transition occur?”
>
> The phase transition will depend on the distribution, so we cannot make a distribution-independent statement of the form you ask for. But, for every distribution, there is some epoch where you are “accurate enough” (in terms of eps-accuracy) to guarantee EF. Past this epoch, things should generally work well. This “many epoch” approach is needed for getting a distribution-free algorithm. That is, we can construct distributions that need arbitrarily many epochs to become EF (for example, if values are iid and very close to a point mass, it may be the case that bundle sizes need to be within some additive delta of each other to get EF).
>
> - “In the model that takes ordinal information as rankings, the preliminaries section does not properly explain how to deal with weak preferences or ties. How does the algorithm deal with ties when eliciting ranks of the fresh item? Can you elaborate on this?”
>
> Since distributions are continuous without point masses, ties are unlikely (zero probability events). So, our algorithms can deal with ties in an arbitrary way (e.g., “if this item is tied with a previous item for any agent, ignore it”) without any changes in the formal guarantees. We will include a clarification in the preliminaries.
>
> - “Updating memory: in Algorithm 2, at each epoch an agent’s memory is updated with comparison with (prior) k^6 items. What is the reason behind such updating mechanism? If an agent’s error is not within the tolerance, an arbitrary item is chosen to be remembered. This seems a bit arbitrary (no pun intended). Is the reason mainly due to assumptions on distributions? Wouldn’t it be more beneficial to remember a certain item (say the one with the average highest ranking)?”
>
> This specific choice is just to make our analysis and algorithm description a bit cleaner. In the analysis, we see that the probability this occurs is so small that, when it happens, it can essentially be handled in any reasonable way. For succinctness, it happened to be more convenient to force the sampling phase length and bound the probability of error rather than continuing to sample and bounding the expected length of sampling. Also as a small note, in our model, we are only allowed to assign an item to memory on arrival, so we can’t go back and choose a reasonable item we’ve seen in the past.
>
> - “Algorithm 1 uses more memory compared to Algorithm 2 but they both achieve the same guarantee on the non-iid model. However, Algorithm 1 offers shorter epochs. Could Algorithm 1’s bound improve with longer epochs? In other words, is the bound tight both for iid and non-iid models?”
>
> The EF guarantee for both algorithms is the same (and can’t be improved asymptotically). Given a distribution and a fixed number of items, if one closely looks at the probability bounds in our proofs, Algorithm 1 will have a smaller probability of not achieving EF (but, of course, for both algorithms this probability is vanishing).
> In terms of efficiency, these are the best bounds we can find. Using our current analysis on longer epochs/more items would not help as we would not be able to bypass the bounds on the “ideal” algorithms which are currently the same; it is plausible that a different analysis can separate the two algorithms in terms of their efficiency guarantees (but still their efficiency is upper bounded by 18/19).
>
> - “The crux of the paper is assuming envyfreeness and devising exploration techniques to achieve reasonable welfare approximations. I wonder whether it is possible to improve on efficiency (or its approximations) by relaxing the fairness notion to EF1, proportionality, etc.? The first question is whether the impossibility results in Theorem 4 or 12 still hold.”
>
> Both theorems are for two agents, therefore the same results hold for proportionality.
>
> Theorem 12 definitely holds if we replace EF with EF1: notice that EF is used in the proof to bound bundle sizes; EF1 (or even weaker notions) would actually give the same bounds (see supplementary material, starting at line 823).
>
> Theorem 4 does not seem to hold immediately, but we believe it should (some other trick would be needed).
>
> - “In general, simultaneously achieving fairness and strategyproofness is a challenging task; and it is often impossible in fair division. I wonder if there is a way to achieve strategyproofness under the settings of this paper (e.g i.i.d bounded or unbounded models)?”
>
> This is a very interesting open problem. It is plausible that uncertainty about the future can help the designer in achieving strategyproofness. To the best of our knowledge, this is open even in the known value setting.

---

> > ### Comment · Reviewer_c1uw · 2022-08-07
> > **Thanks and overall comment**
> >
> > Thank you for addressing my questions. The result in the paper are novel and certainly interesting. I agree with other reviewers that the algorithmic approaches are for the most part straightforward; however, the novelty and challenges often rely on technical proofs. And I believe the proofs are certainly non-trivial and require reasonable technical work.

---

### Official Review · Reviewer_8ye5 · 2022-07-11

**Rating:** 7
**Confidence:** 3
**Soundness:** 3 good
**Presentation:** 3 good
**Contribution:** 3 good

**Summary:**

This paper studies a fair allocation problem in a dynamic setting, where items arrive one by one. Agents have additive valuations and a valuation of an items follows an unknown distribution (i.i.d/non-i.i.d.). In this paper, it is assumed that an algorithm cannot know the cardinal information of the valuation, but can obtain the ordinal information. For the i.i.d. case when agents can compare a new item with any subsets of previously allocated items, the authors give an algorithm whose output is EF and also $(1-\epsilon)$-approximation to optimal utility with high probability. When agents can compare a new item with one of previously allocated items, the authors give an algorithm whose output is EF and also $((1-1/e)^2-\epsilon)$-approximation to optimal utility with high probability. Finally, for the non-i.i.d. case, the authors show that the aforementioned two algorithms output an allocation that is EF and 1/e-approximately Pareto optimal with high probability.

**Questions:**

Is there any relationship between the problem of this paper and the secretary problem? The idea of deciding a threshold by estimation also appears in papers related to the secretary problem.

**Strengths And Weaknesses:**

- Originality: The problem setting where an algorithm has access only to the ordinal information is novel in the line of dynamic settings. The idea of proposed algorithms is to estimate the distribution by sampling, and this is done by combining existing ideas carefully.
- Quality: The results seem correct as far as I have checked. It is better to mention some application if the authors have some in mind. The framework of the proposed algorithms looks like the $\epsilon$-greedy algorithm of the stochastic bandit problem. It may be possible to improve the performance by incorporating ideas of bandit algorithms.
- Clarity: I feel that this paper is mostly well-written. I recommend the authors to place a table that summarizes the contributions (about possibilities and impossibilities) so that the reader can grasp the main results at a glance.
- Significance: I do not think that the fair allocation attracts much attension in the NeurIPS community. Nevertheless, the proposed algorithms is built by a kind of learning technique, and seem closely related to the bandit algorithms, so I guess that this paper is related to this conference.
- Minor comments:
    - The term $\alpha$-Pareto efficient can be changed to $\alpha$-Pareto Optimal for consistency of terms. In Theorem 13, the authors use “$(1/e-\epsilon)$-Pareto optimal.”
    - line 169: $v_i(V_j)$ → $v_i(A_j)$
    - If $c$-strongly-EF was introduced before, it is better to add a reference in the definition.
    - For the non-i.i.d. model, the criteria of efficiency is moved on to the Pareto optimality from the utilitarian welfare. I guess that this is because it is hard to approximate the utilitarian welfare in the non-i.i.d. case. It may be helpful if the authors mention about this in the introduction of Section 6.
    - There are some other recent papers related to the online fair allocation, say,
        - Banerjee, Gkatzelis, Gorokh, and Jin. Online Nash Social Welfare Maximization with Predictions, SODA 2022
        - Kawase and Sumita. Online Max-min Fair Allocation, SAGT 2022

        If the authors find some relationship, please refer these papers.

---

> ### Author Response · Authors · 2022-07-31
> **Response to Reviewer 8ye5**
>
> Thank you for your review and constructive feedback. Below we answer your question.
>
> - “Is there any relationship between the problem of this paper and the secretary problem? The idea of deciding a threshold by estimation also appears in papers related to the secretary problem.”
>
> There is no technical connection to the secretary problem or other problems in optimal stopping theory (e.g. prophet inequalities) as far as we know. Also, see our response to Reviewer rhEt regarding the random order version of our problem.

---

> > ### Comment · Reviewer_8ye5 · 2022-08-08
> > **Thank you for the comment**
> >
> > I have read your reply. I appreciate the additional remark on the adversarial and random order models. Although I have not understood exactly the impossibilities in the random order model, mentioning those points in the main body would strengthen the motivation to consider the iid model.

---

### Official Review · Reviewer_qJFx · 2022-07-13

**Rating:** 5
**Confidence:** 3
**Soundness:** 4 excellent
**Presentation:** 4 excellent
**Contribution:** 3 good

**Summary:**

Fair division of indivisible goods is a flouring research area in computation social choice. There are some works that study the dynamic setting when the items arrive one by one, and the algorithm needs to make immediate and irrevocable decisions at the time when a new item arrives. The existing works already show that when the values are independently drawn from an identical distribution, we can achieve envy-free with high probability and PO simply by allocating each item to the agent who has the highest value. This result can be extended to the cases of agent-specific distributions and correlated agents. All these results require the knowledge of distributions and the realized values, which motivates the current work to study the partial information setting when the algorithm only has ordinal (relative) rankings.

The current work assumes that at the time a new item arrives, the relative rankings of the item in each agent’s bundle are revealed to the algorithm, based on which the algorithm needs to allocate whom to allocate it. Different from the cardinal setting, the authors first showed that envy-freeness with high probability is incompatible with (exact) Pareto optimality, but compatible with approximate PO and utilitarian optimal welfare. The algorithm, consisting of exploration/sampling phase and exploitation/ranking phase, is natural, where the first phase is used to store samples that can be used to calculate the empirical quantiles. The authors (approximately) optimized the length of the sampling length where longer means more accurate empirical quantiles but more inefficient allocations.

Secondly, the authors also studied a more demanding setting where the relative ranking is restricted to a fixed number of items, representing the limited memory situation. Fortunately, the previous results can almost be recovered.

Finally, the authors also explored the extent to which fairness and efficiency are compatible for non-iid models.

Questions:

What if the distributions are not bounded or continuous? How will they change the current results?

Typos:
The paper is well written, as far as I check. I did not find many typos, but \ldots and \cdots should be consistent.

**Questions:**

What if the distributions are not bounded or continuous? How will they change the current results?

**Limitations:**

N.A.

**Strengths And Weaknesses:**

Strengths:

The partial-information model is very interesting to me. The results are natural and complete. The paper is well-written and easy to follow.

Weaknesses:

I worry about the technical contribution. The algorithms are very standard by using the first phase to record the samples which provide empirical quantile estimations for items in the second phase. Of course, the results are not straightforward, but I did not find many technical novelties.

---

> ### Author Response · Authors · 2022-07-31
> **Response to Reviewer qJFx**
>
> Thank you for your review and constructive feedback. Below we answer your question.
>
> - “What if the distributions are not bounded or continuous? How will they change the current results?”
>
> For non-continuous distributions, ties might lead to technical difficulties. For example, non-continuous distributions allow point masses, making EF allocations generally impossible (e.g. the parity of the number of items starts to matter).
> Unbounded distributions also seem difficult to handle. For example, there exist unbounded distributions where every agent *must* get their favorite item, which means that a lot of our asymptotic bounds/sums of independent things do not work out nearly as nicely.

---

### Author Response · Authors · 2022-07-31
**Response to all reviewers**

We thank all reviewers for the thorough reviews and helpful comments. Since there are no concerns/questions that are common across reviewers we will respond to each reviewer’s questions in separate comments. We will incorporate the valuable suggestions from all reviewers in the final version of this paper.

---

### Meta-Review · Area_Chair_ykfb · 2022-08-26

**Recommendation:** Accept
**Confidence:** Certain

**Metareview:**

All reviewers are positive about the paper and found that the problem of sequential/online fair allocation of indivisible items interesting (and relatively new), and the theoretical results significant and sometimes surprising. Technically, the paper also has a "bandit" flavor, which makes it a good fit for NeurIPS.

**Award:**

No

---

### Decision · Program_Chairs · 2022-09-14

Accept